# Conformational distortion in a fibril-forming oligomer arrests alpha-Synuclein fibrillation and minimizes its toxic effects

Ritobrita Chakraborty [1], Sandip Dey[1], Pallabi Sil[3], Simanta Sarani Paul[4], Dipita Bhattacharyya[2], Anirban Bhunia[2], Jayati Sengupta [1✉] & Krishnananda Chattopadhyay [1✉]

The fibrillation pathway of alpha-Synuclein, the causative protein of Parkinson's disease, encompasses transient, heterogeneous oligomeric forms whose structural understanding and link to toxicity are not yet understood. We report that the addition of the physiologically-available small molecule heme at a sub-stoichiometric ratio to either monomeric or aggregated α-Syn, targets a His50 residue critical for fibril-formation and stabilizes the structurally-heterogeneous populations of aggregates into a minimally-toxic oligomeric state. Cryo-EM 3D reconstruction revealed a 'mace'-shaped structure of this monodisperse population of oligomers, which is comparable to a solid-state NMR Greek key-like motif (where the core residues are arranged in parallel in-register sheets with a Greek key topology at the C terminus) that forms the fundamental unit/kernel of protofilaments. Further structural analyses suggest that heme binding induces a distortion in the Greek key-like architecture of the mace oligomers, which impairs their further appending into protofilaments and fibrils. Additionally, our study reports a novel mechanism of prevention as well as reclamation of amyloid fibril formation by blocking an inter-protofilament His50 residue using a small molecule.

[1] Structural Biology and Bioinformatics Division, CSIR-Indian Institute of Chemical Biology, Kolkata, India. [2] Department of Biophysics, Bose Institute-Centenary Campus, P-1/12C.I.T. Scheme VII-M, Kolkata, India. [3] Present address: Department of Physics, University of Alberta, Edmonton, AB, Canada. [4] Present address: Department of Medicine, Centre for Prion and Protein folding disease, University of Alberta, Edmonton, AB, Canada. ✉email: jayati@iicb.res.in; krish@iicb.res.in

A plethora of evidences associate the fibrillation of the intrinsically unfolded protein alpha-Synuclein (α-Syn) with the neurodegenerative motor neuron disorder Parkinson's Disease (PD)[1,2]. PD is characterised by the presence of fibrillar cytosolic Lewy body plaques within the dopaminergic neurons of the substantia nigra pars compacta of the mid-brain. In vitro, α-Syn has been shown to follow the nucleation-conversion-polymerization model of aggregation[3,4]. The kinetics of this pathway initiates with the primary nucleation phase, in which the native monomeric protein misfolds and combines to form 'pre-nucleation clusters' or 'pre-nuclei', which then convert into oligomer seeds (nuclei), that subsequently elongate via monomer addition, to give rise to fibrils. Current hypotheses predict that the initial oligomeric intermediates of fibrillar structures, common to many neurodegenerative disease-related proteins including α-Syn, are the primary toxic species[5]. Upon maturation, the fibrils which are otherwise less toxic[6], fragment into oligomers which act as secondary nuclei or 'seeds' for de novo fibrillation on existing fibril surfaces, giving rise to a feedback loop of fibril amplification[7]. However, a detailed understanding of the structural basis of the pathogenicity of these oligomeric assemblies/ conformations is not available yet owing to their structural heterogeneity.

Several structural descriptions of mature fibrillar assemblies manifest the universality of the amyloid cross-β architecture (wherein β strands align perpendicular to the fibril axis, thereby generating arrays of β sheets that are oriented parallel to the fibril axis)[8]. A solid-state NMR (ss-NMR) study has revealed a 'Greek-key'-like β-sheet topology of α-Syn monomers within a protofibrillar assembly[9]. This description also coincides with the conserved bent β arch 'kernel' architecture that forms the core of each identical protofilaments that constitute the rod and twister fibril polymorphs studied recently[10]. Although 3D structures of fibrils from different proteins like α-Syn, Tau, and amyloid β have recently been solved[10–16], structural understanding of the intermediate oligomers which precede fibrillation is limited[17]. There may be several reasons for this: first, these oligomers are transient with rapid interconversion dynamics, and second, the process of fibrillation manifests through numerous pathways thereby resulting in an apparent low concentration, at any particular timepoint, of the intermediate oligomeric populations. Since there is a necessity in endeavours that target the oligomers in order to combat PD toxicity, a detailed structural validation of these intermediate conformers is essential for a possible therapeutic intervention.

Here, we show that some of the afore-mentioned limitations could be overcome by using a small molecule heme (hemin chloride), which when added to either monomeric or aggregated α-Syn, results in a relatively-homogeneous and stable population of nontoxic tetramers whose structure could subsequently be elucidated using cryo-electron microscopy (cryo-EM). In addition to the obvious importance of the use of heme—which enables for the first time, structural characterization (albeit at a low resolution) of a < 100 kDa nontoxic oligomer within the aggregation pathway of α-Syn, the heme-α-Syn system provides an additional important biological problem corresponding to a number of neurodegenerative diseases including PD. It was previously shown that heme can inhibit aggregation of α-Syn[18,19] by forming distinct annular oligomers similar to those formed in pathogenic mutations of α-Syn. However, the mechanism of inhibition of aggregation as well as whether such oligomers are cytotoxic and can retard further formation of larger oligomeric species and protofilaments was largely unexplored. Ferric dihydroporphyrin IX and related macrocyclic compounds have also been shown to inhibit amyloid fibril formation with $IC_{50}$ values in the low micromolar range[20–22]. Importantly, several heme-proteins have been reported to possess protective functions in neurodegenerative diseases. Cytochrome c is present in abundance within α-Syn Lewy body neurites where the two proteins form an anti-apoptotic, peroxidase activity-positive covalently bonded hetero-oligomer[23,24]. Ferric cytochrome c has also been shown to inhibit α-Syn aggregation[24]. Under physiological conditions within the human brain and peripheral RBCs, the heme cofactor-containing neuronal haemoglobin scavenges α-Syn, leading to a reduction of PD-induced mitochondrial damage and apoptosis[25]. Additionally, overexpression of the hemeprotein neuroglobin inside neuronal cells, reduces cytoplasmic α-Syn inclusions and associated mitochondrial damage[26].

Initial structural characterization of the heme-treated oligomers using FT-IR revealed a reduction in their antiparallel β sheet content, compared to the fibril-forming oligomers that had not been treated with heme. In accordance to previous reports[22,27,28], this finding can be correlated with the reduced transcellular seeding and low pathogenicity, to both bio-membrane-mimicking liposomes and neuroblastoma cells, of the heme-stabilized oligomers. Cryo-electron microscopy (cryo-EM) of the heme-treated, nontoxic oligomers revealed that heme primarily stabilizes a 'mace'-shaped oligomeric structure of α-Syn and thus arrests fibrillation. A Greek-key-like architecture of the elementary unit of a protofilament of α-Syn has been identified by ss-NMR[9,10]. The size and shape of the mace oligomer closely matches with the Greek-key-like model comprising approximately four α-Syn monomers. SEC-MALS, fluorescence correlation spectroscopy and native PAGE data also demonstrated that these heme-treated oligomers are tetrameric. Closer analysis of the heme-stabilized mace oligomer, however, indicated that this structure is likely a distorted version of the Greek-key-like motif. Therefore, the mace oligomer has been termed as the 'twisted' Greek-key-like oligomer. Our findings suggest that heme targets and distorts the Greek-key-like fold of the mace-shaped oligomers thereby preventing their hierarchical assembly into protofilaments and fibrils.

This distortion in the Greek-key-like conformation occurs upon heme binding to the His50 residue located at the inter-protofilament hydrophobic steric zipper interface (conserved residues that form a hydrophobic packing site within the fibril core) within the 'rod' fibril polymorph, which is the predominant fibril polymorph formed in the case of wild-type α-Syn[10]. We show using a His50Gln mutant that the histidine residue is obligatory for the binding of heme to α-Syn, which may weaken the association between His50 of one protofilament and Glu57' of the opposite/adjacent protofilament, preventing the association of the two into a mature fibril. In preformed mature fibrils, heme binding to His50 at the zipper interface destabilizes the protofilament interface integrity leading to an 'unzipping' of the fibril into its constituent protofilaments. Thus, our study offers an explanation of how the structural fold of the mace-like building blocks or oligomers of amyloid fibrils can be modified (distorted) by targeting small molecules to specific residues that maintain fibril integrity, thereby rendering the resulting structures nontoxic. This technique can be exploited for therapeutic intervention of similar amyloid fibril models such as Tau and Amyloid β[12,13] at various stages of their erroneous self-assembly.

## Results and discussion

We observed the impact of heme on the aggregation of α-Syn under two circumstances. In the first condition, heme was added to monomeric α-Syn, after which the reaction mixture was subjected to incubation at 37 °C under constant agitation, a process that shall henceforth be termed as 'pre-incubation' in this manuscript. In the second condition (termed 'post-incubation'), we treated aggregated heterogeneous prefibrillar and fibrillar

conformers (which were generated by 48 h of aggregation) with heme. We found that when heme was added to α-Syn at a sub-stoichiometric molar ratio of α-Syn/heme 25:1, at either of the two afore-mentioned circumstances, a population of small, morphologically homogeneous oligomeric structures were formed. The oligomers formed due to preincubation of α-Syn with heme have been named 'oligomers$_1$' while those that are formed as a result of postincubation have been labelled 'oligomers$_2$' (Supplementary Fig. 1, Supplementary Information or SI).

**Heme inhibits fibril formation, seeding, and fibril maintenance within the aggregation pathway of α-Syn.** We used Thioflavin T (ThT) fluorescence to investigate the effect of heme on the different events that occur at the molecular level within the amyloid assembly process of α-Syn. In the absence of heme, we observed the distinctive sigmoidal aggregation kinetics[29,30] of α-Syn (Fig. 1a). The ThT data were complimented by imaging using negative-stain TEM and AFM, which clearly showed the formation of a fibrillar network of α-Syn (Fig. 1b). The diameter of the mature fibrils after 96 hours of aggregation typically varied between 8.5 and 10 nm (TEM data, Fig. 1b), while the mean height measured 9.7 nm (AFM topographic data, Fig. 1b). The length of the fibrils measured between 0.5–2 μm. Under the condition of preincubation with heme, the protein showed

insignificant ThT fluorescence, indicating inhibition of fibril formation and elongation, in a dose-dependent manner (Fig. 1a). The ThT data were supported by the negative-stain TEM and AFM results, which showed that upon preincubation with heme, fibril formation did not occur even after 96 h of incubation (Fig. 1c). Instead, the aggregation was stopped at the oligomer$_1$ stage, the diameters of which varied between 1 and 3.5 nm, while the mean height according to AFM measurements was 2.3 nm (Fig. 1c).

α-Syn aggregation is known to sustain via toxic protein templating or seeding, wherein, oligomers consisting of misfolded monomers as well as those oligomers, which are formed from the fragmentation of fibrils (and are called seeds) characteristically induce monomeric forms to oligomerize and fibrillate. We compared this propensity of heme-treated and untreated seeds by treating α-Syn monomers with them under aggregation-inducing conditions. Seeds were initially produced by incubating α-Syn for 96 h under aggregation-inducing conditions in the presence or absence of heme, which was followed by sonication. These heme-treated and untreated seeds were then added to fresh α-Syn monomers, which were allowed to aggregate. In contrast to the usual de novo sigmoidal aggregation behaviour of α-Syn, the addition of fibril seeds to the monomeric protein followed by induction of conditions that favour aggregation (i.e., 37 °C and constant agitation) resulted in hyperbolic aggregation kinetics

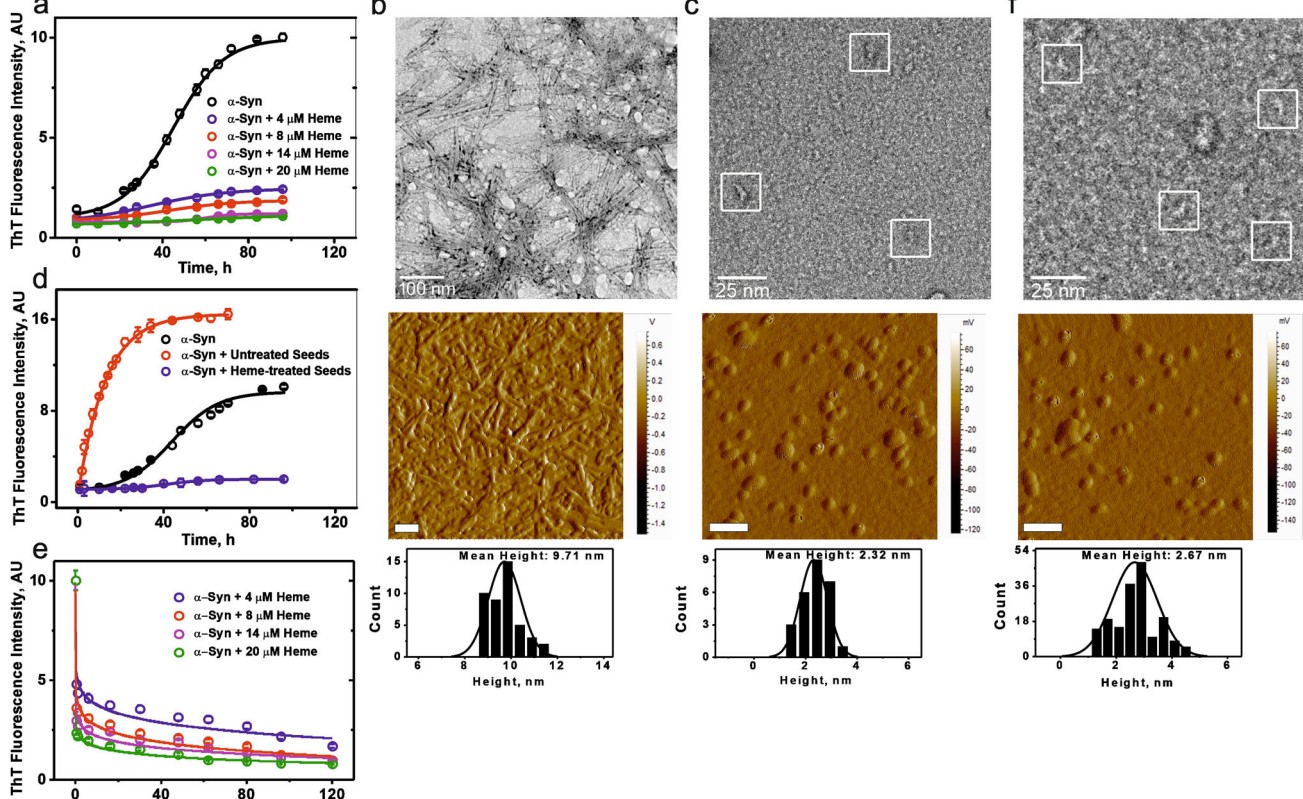

**Fig. 1 Heme arrests fibrillation of α-Syn by converting the heterogeneous mass of aggregates into uniform-sized oligomers. a** Dose-dependent study (ThT fluorescence) of 200 μM monomeric α-Syn preincubated for 96 h with heme. **b, c** Negative-stain TEM and AFM micrographs of **b** α-Syn fibrillar network formed in the absence of heme after 96 h of aggregation; **c** α-Syn oligomers$_1$ (white squares, TEM) formed after 96 h incubation with heme under aggregation-inducing conditions (preincubation). **d** Comparison of the seeding effect of heme-treated seeds (in blue) and untreated seeds (in red) on monomeric α-Syn; the sigmoidal aggregation kinetics profile of 200 μM α-Syn, without any seeds added (in black) has been added for comparison. **e** Dose-dependent study (ThT fluorescence) of the disaggregation of 96 h fibrils upon addition of heme (postincubation). **f** negative-stain TEM (white squares) and AFM micrographs depict formation of oligomers$_2$ upon disaggregation by heme of 96 h fibrils. Error bars in ThT experiments represent ±SEM and have been calculated from eight independent experiments. The scale bars in the AFM micrographs represent 100 nm. The height distribution histograms are presented beneath their corresponding AFM image. The ratio of α-Syn/heme was maintained at 25:1 for all of the above studies.

(Fig. 1d). However, seeds prepared from the protein that had been incubated in presence of heme and then sonicated, appeared to be inert and completely inhibited the seed-induced aggregation kinetics (Fig. 1d).

Furthermore, we observed the impact of heme on mature fibrils, by post-incubating fibrillar structures formed after 96 h of aggregation with heme. A dose-dependent and steady reduction in ThT fluorescence was observed (Fig. 1e), suggestive of fibril breakdown. Furthermore, the ThT intensity remained nominal even after 120 h of heme addition to the fibrils (due to the formation of oligomers$_2$, refer AFM image, Fig. 1f). This observation suggests that heme-induced oligomers$_2$ are unreactive and cannot act as seeds that can grow into new fibrils on preformed fibril surfaces. The negative-stain TEM image (Fig. 1f) depicting the breakdown of the 96 h aggregates showed the presence of a population of oligomers upon incubation with heme.

The mean height of 2.7 nm of the oligomers$_2$ was calculated from the corresponding AFM image (Fig. 1f). Although the AFM and negative-stain TEM data complement the ThT data, we performed a control study to observe if there was any quenching of the ThT dye by heme. Supplementary Fig. 2 shows that increasing concentrations of heme do not have a quenching effect on ThT fluorescence. Collectively, these results demonstrate that heme can arrest fibrillation and convert the heterogeneous aggregates into an inert population of predominantly oligomeric species, or oligomers$_2$.

**Heme reduces in-cell seeding capacity and toxicity of α-Syn oligomers$_1$ and oligomers$_2$.** Within cellular models, fibril-forming α-Syn aggregates have been shown to spread transcellularly and seed de novo aggregation via prion-like mechanisms[31,32]. To demonstrate the nonseeding, nonfibril forming nature of the heme-stabilised structures within a cellular model, untreated as well as heme-treated sonicated fibril seeds (refer Methods) were added exogenously to SH-SY5Y neuroblastoma cells that had been previously transiently transfected with an EGFP-α-Syn construct. Figure 2a shows the confocal image of cells transfected with the EGFP construct, that were subsequently transduced with 1 μM sonicated fibril seeds for 24 h. At this concentration, the majority of cells displayed punctate cytoplasmic inclusions (magnified image in inset). Importantly, transfected cells further transduced with heme-treated sonicated seeds (Fig. 2b) or those transfected with α-Syn-EGFP but not transduced with any seeds (vehicle, Fig. 2c) did not show any fluorescent puncta within their cytoplasm. The cells in Fig. 2b, c show some fluorescence associated with the cell membrane, which is a result of the binding of monomeric α-Syn onto the membrane as has been observed before[33].

Subsequently, we compared the toxicity (on synthetic membranes and neuroblastoma cells) of the structural forms of α-Syn formed at various timepoints (between 0 h and 96 h) during aggregation, in the absence and presence of heme, with the protein aliquots collected every 12 h from the aggregation reactions and added to the membranes/cells. To quantify membrane permeation propensities of the various structural species, we treated calcein-loaded small unilamellar vesicles (SUVs composed of 3: 7 POPC: DOPS, refer Methods for details) with (i) α-Syn set to aggregate at 37 °C under constant agitation, in the absence of heme, (Fig. 2d) (ii) α-Syn oligomers$_1$ formed when the protein is preincubated with heme (Fig. 2e) as well as (iii) α-Syn oligomers$_2$ formed when the protein is postincubated with heme (Fig. 2f). We also added these structural species formed at identical conditions to SH-SY5Y neuroblastoma cells and used FITC-conjugated Annexin V[34] as a marker for the exposed plasma membrane lipid phosphatidylserine (PS), which

is a diagnostic feature of cells undergoing early apoptosis. We used propidium iodide, on the other hand, to detect loss of membrane integrity as a result of late apoptotic and/or necrosis[35] in these cells.

Addition of the heterogeneous population of oligomers, protofilaments and short-length protofibrils, (as observed from AFM measurements, refer Supplementary Fig. 4, SI) that are formed between 24 h and 60 h, on SUVs led to pore formation and release of the fluorescent probe calcein, the extent of which indicated the toxicity of the protein species (Fig. 2d). Similarly, addition of these untreated structural species to SH-SY5Y neuroblastoma cells led to early as well as late apoptosis/necrosis (Fig. 2d). However, when these structures were stabilised using heme to form oligomers$_1$ (preincubated with heme) and oligomers$_2$ (postincubated with heme) showed comparatively reduced toxicity at the equivalent timespan between 24 and 60 h (Fig. 2e, f).

**Structural characterization of the heme-stabilized oligomers reveals their mace-shaped tetrameric assembly.** The aggregation of α-Syn proceeds through multiple pathways, all of which culminate into the formation of fibrils. One of these pathways entails the generation (after 20–24 h of aggregation) of fibril-forming annular oligomers, which are cytotoxic and possess a high content of antiparallel β sheet structure[17]. In order to gather structural insights on the heme-induced prevention of fibrillation and membrane/ cellular toxicity of α-Syn oligomers, we compared the structure of these toxic, annular, fibril-forming oligomers with that of the nontoxic heme-treated oligomers$_1$ and oligomers$_2$.

FT-IR measurements were used to compare the secondary structures[36] of the untreated oligomers with the heme-treated oligomers$_1$ and oligomers$_2$. The FT-IR fitting strategy involved the following: first, the second derivative of the FT-IR spectrum was used for peak position determination as this method enables the separation of overlapping peaks. The double derivative spectra of the untreated oligomers, as well as oligomers$_1$ and oligomers$_2$ are provided in Supplementary Fig. 5, SI show the decrease in the high wavenumber peaks in presence of heme, especially the complete disappearance of the peak at ~1695 cm$^{-1}$. The IR spectra were then deconvoluted based on these peaks identified by the double derivatives. The oligomers formed after 24 h of incubation in the absence of heme (that also showed augmented cytotoxicity, Fig. 2d), contained a large extent (~12%) of antiparallel β sheet structure (Fig. 3a, and Supplementary Table 1, SI) evidenced by high frequency bands between ~1683 and 95 cm$^{-1}$. The low frequency component between 1620 and 38 cm$^{-1}$ was observed to be four times prominent than the high frequency bands, as is expected for antiparallel β sheet structure[17]. In contrast, the oligomers$_1$ (Fig. 3b) and oligomers$_2$ (Fig. 3c) contained a much reduced antiparallel β sheet component (2% and ~5%, respectively, refer Supplementary Table 1, SI). Similar β sheet orientation in the oligomeric species of α-Syn have been reported previously, and oligomer populations with differing characteristics have been shown to cause difference in the membrane perturbation and cell death via apoptosis[27,28]. Thus, heme leads to the conversion of antiparallel β sheet-containing oligomers into parallel β sheet-containing ones, thereby easing the process of conversion of such oligomers into the parallel β fibrils, which are less toxic than these oligomers.

Subsequently, we used cryo-EM in conjunction with single particle reconstruction technique[37] to characterize the 3D structures of heme-treated α-Syn oligomers$_1$ and oligomers$_2$ formed in the presence of heme. Negative stained TEM imaging showed (Fig. 1c: oligomers$_1$ & f: oligomers$_2$, inset of

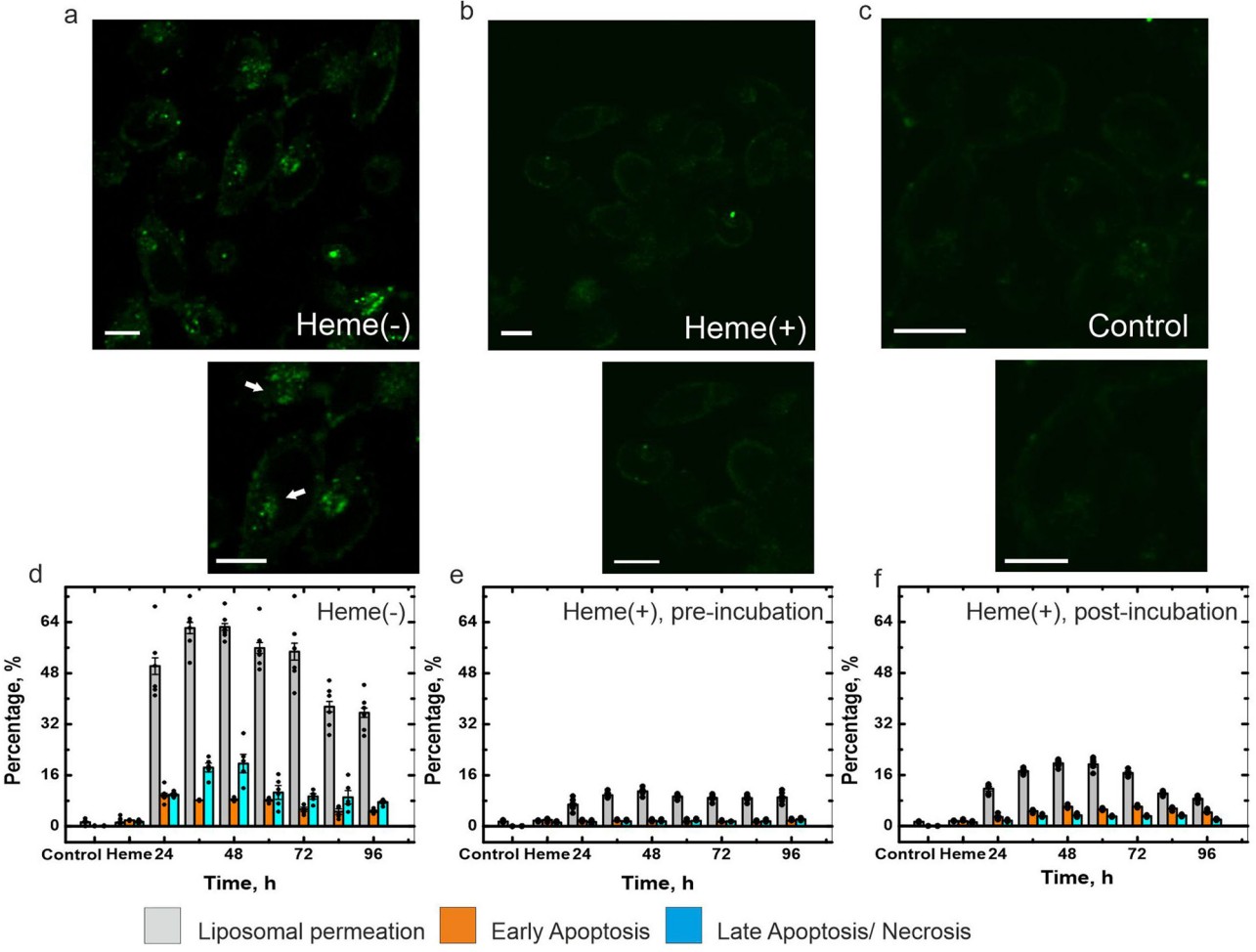

**Fig. 2 Heme minimizes the in-cell seeding, synthetic SUV permeabilization and cytotoxicity of various aggregated species of α-Syn. a–c** α-Syn-EGFP-transfected SH-SY5Y cells were transduced and incubated for 24 h with 1 μM fibril seeds prepared by sonicating preformed α-Syn fibrils that had been incubated under aggregation-inducing conditions **a** without or **b** with heme for 96 h (α-Syn/heme = 25:1). Punctate green structures (white arrows) within the cytoplasm in (**a**, bottom inset) denote seeded α-Syn aggregates. **c** Cells transfected with α-Syn-EGFP but not transduced with seeds (vehicle) do not convert into an inclusion-positive state. The scale bars denote 10 μm. **d–f** The effect of α-Syn structural conformers formed after various periods of aggregation **d** in the absence of heme, or **e** in presence of heme: where heme was added from the beginning of aggregation period (preincubation), or **f** when heme was added after various periods of aggregation had occurred (postincubation), on calcein-loaded liposomal SUV permeation (black bars), early apoptosis (red bars) and late apoptosis/ necrosis (blue bars) as observed in SH-SY5Y neuroblastoma cells. For the FACS and calcein release experiments, error bars represent standard error of the mean and have been calculated from six independent experiments, with each condition replicated thrice.

Supplementary Fig. 3, SI: enlarged view of a single particle) a nearly homogeneous distribution of small linear-shaped oligomers (<10 nm) having a distinct shape, following heme treatment at either preincubation or postincubation stage. The cryo-EM micrographs depicting the oligomers$_1$ (Fig. 4a) revealed a predominant distribution of small particles of uniform shape and size (~6–8 nm). Despite the small sizes, the particles were visually identifiable (presumably heme binding to the protein oligomer produces additional contrast). Particle picking, and initial model building were performed using EMAN2[38]. Reference-free two-dimensional (2D) classi-fications of the particles were performed by the image processing programs Xmipp, RELION and EMAN2 (Fig. 4b: I, II, and III, respectively). Final reference-based (using the EMAN2 initial model) 3D reconstruction and refinement were done in SPIDER[39] (Fig. 4c). To ensure the reliability of the map, we used several validation strategies (refer Methods and Supplementary Figs. 6, 7B, 7D and 7F, SI).

The cryo-EM micrographs depicting the oligomers$_2$ in contrast, showed an ensemble of particles of different sizes along with a few short-length fibrillar structures (Fig. 4d). We selected the population of oligomers$_2$ that appeared qualitatively similar in size to the particles observed as oligomers$_1$ (~6–8 nm; Fig. 4d). These small-sized particles were selectively picked, following which three reference-free 2D analyses were performed (Fig. 4e), and a 3D cryo-EM map was generated following the same procedure used for oligomers$_1$ (Fig. 4f). The 3D cryo-EM maps (data processed independently) of both the oligomers$_1$ and oligomers$_2$ (Fig. 4c, f) appeared as a club or a mace with a heavy head (we termed this structure of the oligomers$_1$ and oligomers$_2$ as the 'mace oligomer'). The resolutions of the maps generated from dataset of oligomers$_1$ and oligomers$_2$ were estimated to be 10.4 Å and 8.9 Å, respectively, at the 0.143 FSC criterion (Supplementary Fig. 7A and 7C, SI).

**Heme distorts the tetrameric Greek-key-like architecture by interacting with His50.** We then investigated the binding affinity and binding site of heme on tetramethylrhodamine-5-maleimide (TMR)-tagged monomeric α-Syn Gly132Cys (G132C) mutant, by

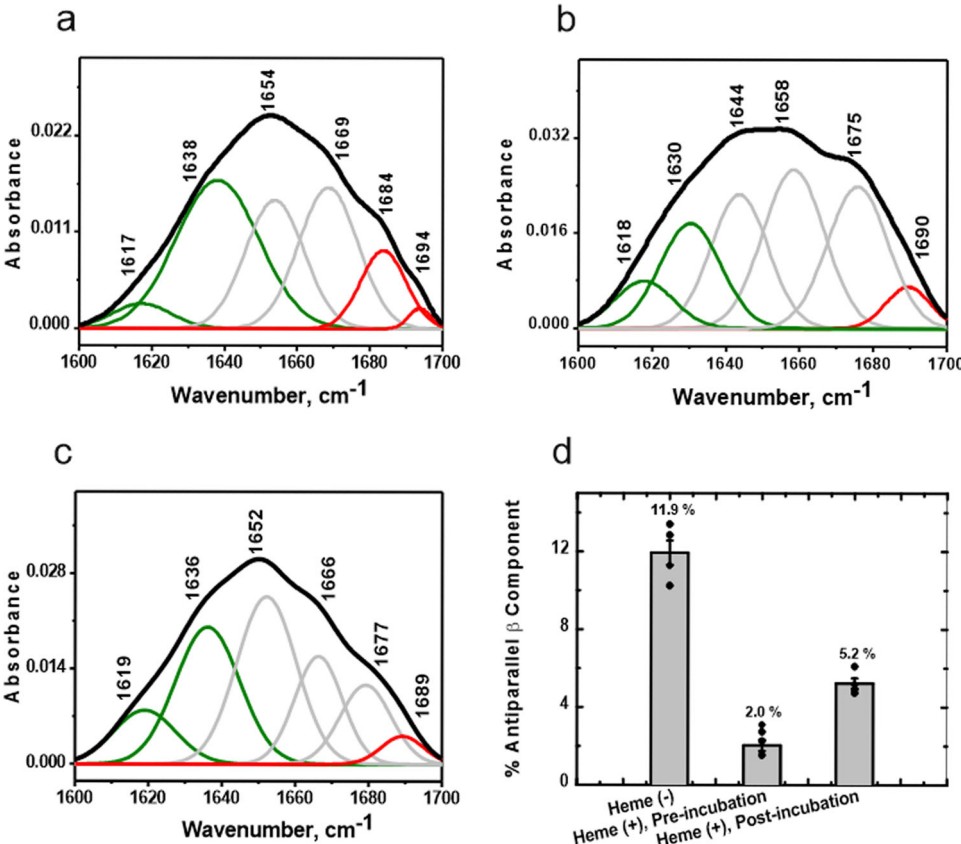

**Fig. 3 Heme causes a major reduction in the antiparallel β sheet component of fibril-forming aggregates.** Deconvoluted FT-IR spectra of 24 h oligomers incubated **a** in the absence of heme; or **b** in presence of heme from the beginning of incubation period (oligomers₁, preincubation); **c** depicts the IR spectrum of oligomers₂ formed when 48 h prefibrillar and fibrillar aggregates were treated with heme, postincubation. **d** Addition of heme causes a drastic reduction in the antiparallel β sheet component (bands at ~1680–90 cm⁻¹, Panels **a**–**c**, illustrated in red) of the resulting oligomeric population. The bands depicted in green denote the parallel β sheet component (1622–1638 cm⁻¹). Error bars are ±SEM obtained from four independent experiments. Second derivatives of the FT-IR spectra (Supplementary Fig. 5, SI) have been used to determine individual peaks.

studying the heme-induced quenching of TMR emission intensity. The cysteine insertion was needed as this residue is obligatory for the maleimide labelling. This single cysteine mutant is considered similar to the wild-type protein for biophysical studies as has been reported previously[29,40]. While the dissociation constant ($K_d$) and number of binding sites on the α-Syn-G132C monomer for heme was 0.59 μM and 1, respectively (Fig. 5a), heme binding to α-Syn was not observed in presence of excess (10 mM) imidazole (Supplementary Fig. 12, SI), suggesting a probability that binding of the heme takes place at a histidine residue in the protein.

A molecular modelling exercise of the heme-induced mace oligomer structure suggested His50, Tyr39, Tyr125, Tyr133, and Tyr136 as five potential residues, which are present on the solvent-exposed surface of the mace oligomer structure and could be involved in the oligomer-heme binding and stabilization (Fig. 5b). Heme-bound α-Syn was also found to have a considerably high peroxidase activity[41], compared to free heme or unbound α-Syn (Fig. 5c). Since His50 is the only histidine positioned at this region, we prepared a His50Gln (H50Q) mutant. Although H50Q aggregated to an extent similar to the WT protein, its aggregation was not inhibited by the addition of heme (Fig. 5d: ThT; 5E-F: AFM). Moreover, the H50Q mutant did not show any binding to heme as observed from the fluorescence binding measurement (Fig. 5G). Binding studies using fluorescence were complimented by using 1D NMR (Supplementary Fig. 13, SI), which also showed no significant

heme binding to the H50Q mutant. Interestingly, H50Q+ heme had a similar peroxidase activity as free heme (Fig. 5c). These results confirmed the crucial role of His50 in heme binding which prevents fibrillation by converting the aggregated protein mass into a population of mace oligomers. Incidentally, H50Q is a missense mutation that causes the late-onset familial form of PD and dementia[42]. H50Q has been recently shown to form a major class of 'narrow' protofilaments, which are 5 nm wide (and a very small class of 10 nm diameter 'wider' fibrils) as it aggregates. Based on our data, we propose that these narrow protofilaments cannot join to form mature 'wide' fibrils as their steric zipper interfaces are weakened due to the lack of the His50-Glu57' (Glu57' of the opposite protofilament) salt bridge. Also, the absence of the only His residue that can bind to heme and thus salvage fibrillation, is presumably responsible for its susceptibility to fibrillation.

We then used SEC-MALS (Supplementary Fig. 8, SI) to estimate the approximate molecular weight of the heme-treated oligomers. The data from light scattering and differential refractive index detectors reveal the molar mass for the oligomer₁ as 60.2 kDa. This was compared with BSA (control) that showed a molar mass of 66.4 kDa. The molar mass of monomeric protein is estimated to be 14.46 kDa using SEC-MALS, as has been established previously[43]. The tetrameric structure of α-Syn previously studied[44] was for 'non fibrillating native structure' and the proposed fold was α-helical. However, our oligomers are 'non-fibrillating' because of the presence of heme. Native

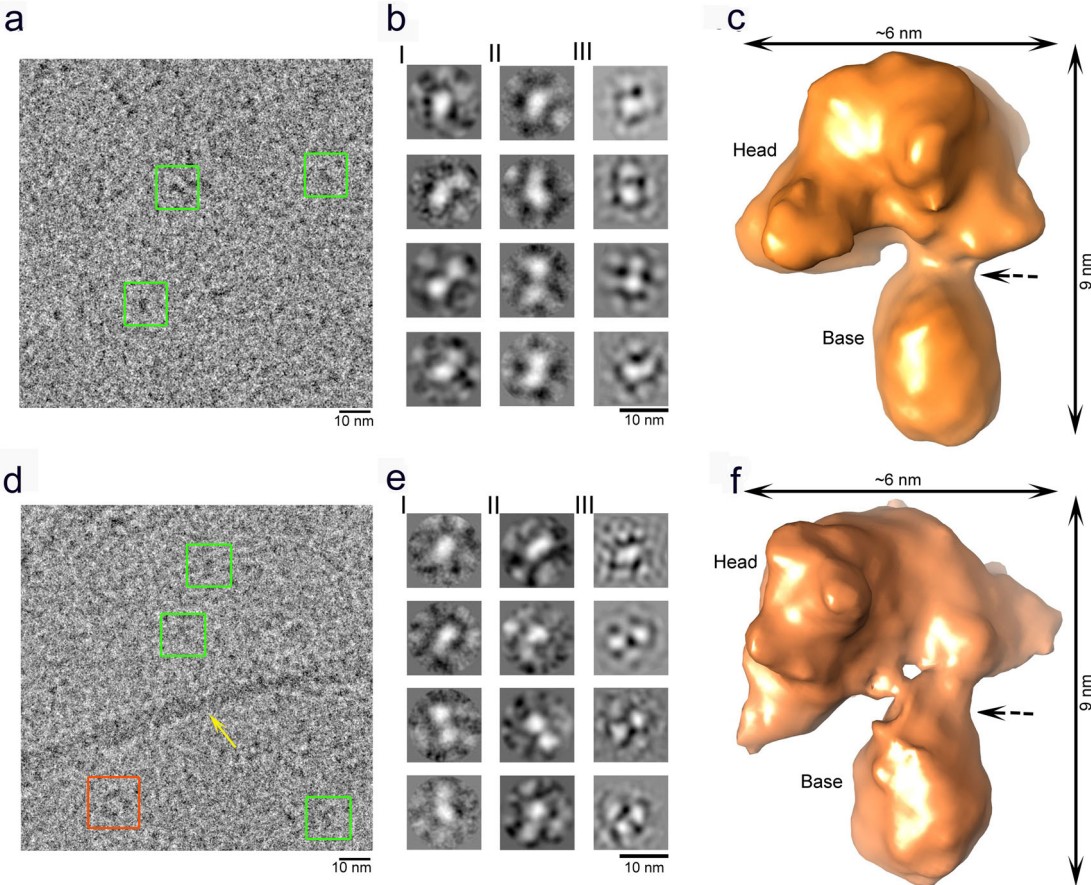

**Fig. 4 Cryo-EM study of heme-stabilized α-Syn oligomers. a–c** Heme treatment on monomeric α-Syn for 24 h (preincubation). **a** Raw micrograph showing uniformly distributed small oligomers (green squares, oligomers$_1$). **b** I–III Reference-free 2D class averaging analyses were done by Xmipp, RELION and EMAN2 (refer Methods). **c** Cryo-EM density map of α-Syn oligomers$_1$. **d, e** Heme treatment performed after fibril formation has initiated, i.e., after 48 h of aggregation (postincubation). **d** Raw micrograph showing distribution of small oligomers (oligomers$_2$) similar to oligomers$_1$, as well as semi-annular-shaped oligomers and fibrils are marked by green squares, orange squares and a yellow arrow, respectively. **e** I–III Reference-free 2D class averaging analyses were done by Xmipp, RELION and EMAN2. **f** The density map of α-Syn oligomers$_2$. In panels **c** and **f**, the dotted black arrow indicates the kink between the head and base of the oligomers.

(nondenaturing) polyacrylamide gel electrophoresis (PAGE, Supplementary Fig. 9, SI) of oligomers$_1$ and oligomers$_2$ (lanes 3 and 4, respectively) showed their migration at a MW of ~60 kDa, while the monomer migrated at ~15 kDa as has been observed before[17,45,46] thereby supporting the SEC-MALS data. These findings were further validated by comparing the theoretical hydrodynamic radius ($r_{H\ Theo}$) of tetrameric α-Syn (prepared from PDB 2N0A)[9] using HullRad[47] (refer Methods), with the experimental hydrodynamic radius of the heme-stabilized oligomers ($r_{H\ Exp}$) determined using fluorescence correlation spectroscopy (FCS, Supplementary Fig. 10, SI). FCS experiments provided experimental estimates of $r_{H\ Exp}$ (5.87 nm and 6.1 nm for oligomer$_1$ and oligomer$_2$, respectively) which were found similar to the theoretical estimate from a tetrameric oligomer ($r_{H\ Theo}$ 6.1 nm). Lastly, the molecular mass (~52 kDa) of the mace-like density map (at the threshold value shown in Supplementary Fig. 7), theoretically estimated from the size of the enclosed volume appeared to be equivalent to ~4 α-Syn molecules (considering that the unstructured N- and C-terminal are only partially visible in the density maps).

When compared, the density maps of both oligomers$_1$ and oligomers$_2$ show a striking resemblance with a tetrameric model of α-Syn arranged in accordance with the Greek-key pattern (2N0A; residues 37–99, excluding the unstructured N- and C-termini) (Fig. 6a)[48]. However, we failed to dock the

entire tetrameric atomic structure into the cryo-EM envelope of the mace oligomer. Hence, we fitted the head and base regions of the tetrameric Greek-key-like motif separately into the mace oligomer density. We found that although the C-terminal structured part of the Greek-key architecture fit well within the head region of the mace oligomer density, a distortion was observed at the junction of the head and the base of the heme-stabilized mace oligomer (Movie 1, SI) giving rise to the formation of the 'twisted' Greek-key oligomer (Fig. 6b, c). Interestingly, the His50 residues (of each α-Syn monomer within the tetrameric mace oligomer) are lined up at the junction of head and base (Fig. 6b) where the distortion is observed. We hypothesize that heme binding to the His50 residues located on the exposed surface at the head-base junction of the tetrameric Greek-key-like structure of the mace oligomer induces a torsion in its topology. To further supplement this data, we performed a molecular docking study followed by molecular dynamic simulations on a tetrameric model of the published PDB 2N0A structure to observe any changes caused by heme. The result of our simulation studies is represented in Supplementary Fig. 14, SI, where (a) shows the well-equilibrated and stable end-simulation structure of the tetramer with the heme molecule ligated at His50. Figure (b) shows the superimposition of the end-simulation structure onto our cryo-EM fitted model from Fig. 6.

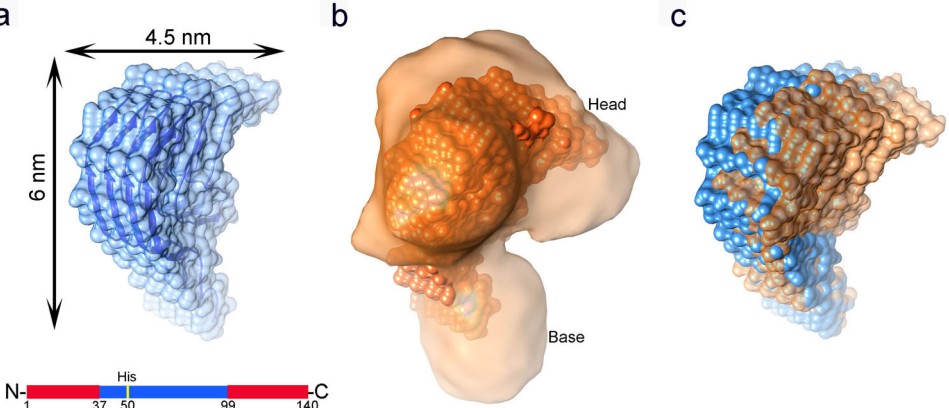

**Fig. 5 Heme binds to the His50 residue on α-Syn forming a peroxidase-positive complex. a** Stern-Volmer plot and double log plot (inset) obtained from the steady-state quenching and binding of TMR-5-maleimide-tagged α-Syn-G132C with heme, $K_D = 0.591\,\mu M$ (datasets = 6). **b** Greek-key β-sheet alignment of the fibril core with the potential heme-binding residues (Tyr: in red; His: in yellow). Inset shows magnified view. **c** Kinetic traces for peroxidase activity monitored for the α-Syn WT-heme complex (datasets = 3). **d** The aggregation behaviour of the histidine mutant H50Q is unaltered in presence of heme as observed using ThT fluorescence. **e–f** AFM micrographs depict that H50Q aggregates into fibrils even in the presence of heme. Scale bars represent 500 nm. **g** The H50Q mutant shows no binding with heme, as observed from the absence of any quenching of TMR-tagged H50Q/G132C by heme (datasets = 3). All error bars are ±SEM.

**Fig. 6 Structural characterization of the molecular mace density map. a** A representation of four α-Syn chains arranged as per the Greek-key architecture (**PDB 2N0A**) in cartoon representation (blue) inside the semi-transparent surface. Only the structured part (residue 37–99) is shown and the unstructured regions (residues 1–36 and 100–140, shown below) are not included in the structure. **b** The full structure shown in figure A could not be fitted into the density map of oligomer$_1$ as a rigid piece. The 'head' and 'base' parts of figure A was fitted separately (orange CPK sphere representations) into the density map of oligomer$_1$ generated from oligomers$_1$ (semi-transparent orange). **c** Superimposition of the Greek-key structure from panel **a** (in blue) and the fitted model from panel **b** (in orange) represented in spheres, shows distortion in the fitted model near its 'head'–'base' junction.

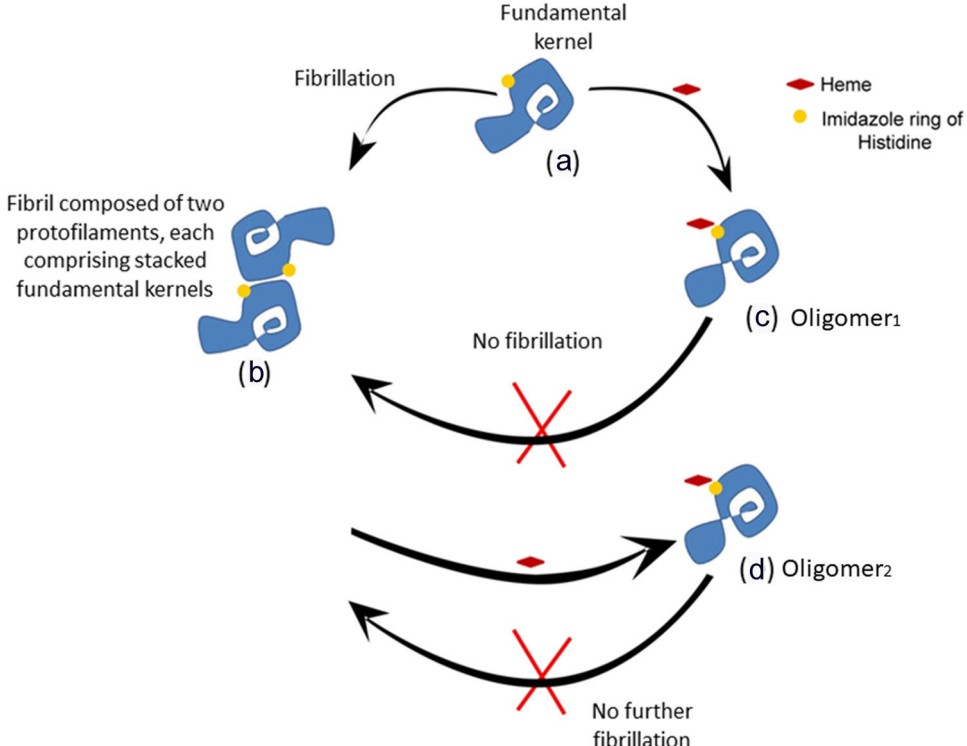

**Fig. 7 Schematic representation of the mechanism of heme-mediated inhibition of α-Syn fibrillation. a** The fibrillation of α-Syn[54,79] proceeds through the formation of an on-pathway (fibril-forming) fundamental protofilament kernel[10] which contains the β sheet Greek-key motif[9] (also termed as a bent β-arch structure) and serves as a 'building block' for fibrillation. **b** Fibril formation occurs via stacking of these kernels into protofilaments which subsequently interlace at the steric zipper interface[10] to form mature fibrils. **c** Addition of heme at an initial stage stabilizes it into a twisted form (oligomer$_1$) which does not form fibrillar structures. **d** Addition of heme at a later stage (when fibrillation has already initiated) causes disassembly of α-Syn oligomers/fibrils by inducing a twist/ distortion in the fundamental kernels resulting in a twisted Greek-key structural fold (oligomer$_2$).

## Conclusion

This study provides a structural insight into the the inhibition of de novo fibrillation as well as breakdown of preformed mature fibrils of α-Syn due to the targeting of an interprotofilament steric zipper interface His50 residue by a small molecule, heme. We suggest the following two possibilities for the mechanism of inhibition: first, heme binding to His50 interferes with the interprotofilament association that stabilizes fibril integrity. Thus, this interaction forces apart the two protofilaments that form the dimer interface (a 'wedge effect' described recently, to investigate fibrillation-inhibition using the polyphenol EGCG[49]). This mechanism is further supported by the recent finding that majority of the fibrils formed from the H50Q mutant are narrow (5 nm wide) due their lack of the His50 residue[50]. Our study therefore implicates a steric zipper residue which can be targeted by drug-like molecules to inhibit fibrillation and its associated toxicity. While previous ideas of using small molecules to target fibrils have been largely restricted to computational docking studies[51–53], recent structural elucidation of fibrillar forms[10,14] of α-Syn have enabled the informed screening and design of small molecules with pharmacophore-like properties. The second mechanism of heme-induced fibrillation-inhibition entails a structural contortion in the C-terminal of the protein assembly (Fig. 6), which is formed upon heme binding to the His50 residue. This contortion prevents the assemblage of the building blocks into protofilaments and fibrils (Fig. 7c). On the other hand, when heme is added to mature fibrils, the structural distortion as a result of heme binding to His50 causes the fibrils disintegrate (Fig. 7d).

The reasons supporting the candidacy of heme as a pharmacophore comprise its endogenous and extensive presence within the human physiological environment, the biological relevance of the hemeprotein-α-Syn system[23,24], the binding efficiency of heme to α-Syn (with a $K_D$ in the nanomolar range) as well as its nontoxicity at the range of concentrations used in this study. Importantly, in this study, the addition of heme aids the conversion of the heterogeneous landscape of fibril assembly into a stable and approximately monodisperse population of oligomers, which has enabled the first ever structural elucidation of nontoxic intermediate oligomers using cryo-EM.

Some groups have identified a predominantly α-helical, native, stable nontoxic tetramer obtained from the nondenaturing purification from cellular sources and brain-derived samples[44,54]. We have been able to demonstrate, using a multitude of techniques including SEC-MALS and FT-IR, that our heme-stabilized oligomer is a β sheet-containing tetramer with a MW of ~60 kDa (MW of α-Syn monomer = ~15 kDa). In summary, this study (at nanometre resolution) of heme bound to α-Syn illustrates how a hydrophobic molecule that nestles between protofilaments within the equally hydrophobic core of amyloid fibrils can weaken interactions that otherwise stabilize fibrils. Efforts are underway in our laboratories to develop effective biologically-functional mechanisms of heme transmission to α-Syn aggregates within animal model systems.

## Methods

For the purification of α-Syn, Tris salt, isopropyl β-D-galactopyranoside (IPTG), phenylmethanesulfonylfluoride (PMSF), ammonium sulfate, urea, sodium dihydrogen sulfate, disodium hydrogen sulfate and sodium chloride from Sigma–Aldrich (St. Loius, MO, USA) were used. Thioflavin T, calcein and hemin chloride were purchased from Sigma–Aldrich. Glycerol and sodium dodecyl sulfate (SDS) were purchased from Merck (Kenilworth, NJ, USA) and USB (Cleveland, OH, USA), respectively. For high performance liquid chromatography (HPLC),

two columns from Waters Corporation (Milford, MA, USA) were used. For the liposome permeation study, the lipids were obtained from Avanti Polar Lipids (Alabaster, AL, USA). FBS, DMEM, Opti-MEM were obtained from Life Technologies (Carlsbad, CA, USA). Cell culture media, supplements, kits and reagents were purchased from Invitrogen (Carlsbad, CA, USA), unless specified otherwise. The dyes Alexa Fluor 488 $C_5$ maleimide and Tetramethylrhodamine-5-maleimide were purchased from Thermo Fisher (Waltham, MA, USA) and Invitrogen, respectively. All other chemicals used for this study were obtained in the highest grade available.

**Expression and purification of α-Syn protein**. Recombinant human α-Syn wild-type (WT), G132C and H50Q/G132C mutants were expressed in *Escherichia coli* BL21 (DE3) strain transformed with the pRK172 α-Syn wild-type or mutant plasmid. Site-directed mutagenesis was performed using the Agilent Quikchange Lightning site-directed mutagenesis kit (Agilent Technologies, Santa Clara, CA, USA). The expression of the protein was induced by adding 1 mM IPTG to a culture that had reached an $OD_{600}$ of 0.5–0.6. The cultures were then incubated at 37 °C with shaking at 180 rpm for 4 h. Cells were harvested by centrifugation. The cell pellets were resuspended in sonication buffer (20 mM Tris, pH 7.4, 100 mM NaCl and 1 mM PMSF) and lysed by sonication using short but continuous pulses at 12 Hz for 1 min. This step was repeated 14 times to lyse all the cells. The lysate was centrifuged at 14,000 rpm for 45 min at 4 °C to remove cell debris. The lysis suspension was brought to 30% saturation with ammonium sulfate and the pellet was discarded. This was followed by 50% saturation with ammonium sulfate. The solution was then centrifuged at 20,000 rpm for 1 h at 4 °C. The resultant pellet was dissolved in 20 mM Tris buffer, pH 7.4 and dialyzed overnight against the same buffer. After dialysis, the protein sample was filtered using a 30 kDa centricon filter (Merck Millipore, Darmstadt, Germany). The crude protein was then injected into a DEAE anion exchange column equilibrated with 20 mM Tris (pH 7.4) and eluted using a NaCl gradient. α-Syn was found to be eluted at about 300 mM NaCl. Fractions containing α-Syn (analyzed by Coomassie-stained SDS-PAGE) were concentrated and further purified using a Sephadex gel filtration column. Fractions containing purified α-Syn were combined and lyophilized. The protein was determined to be about 95% pure by SDS-PAGE. For the sample preparations of all experiments, lyophilized protein was dissolved in 20 mM sodium phosphate buffer (pH 7.4) and filtered using 0.22 μm low protein binding membranes (Millex-GP, Merck Millipore, Germany). The protein concentration was determined by the measurement of absorbance at 277 nm using the extinction coefficient 5960 cm$^{-1}$ M$^{-1}$.

**Preparation of heme-treated oligomers$_1$ and oligomers$_2$, and untreated on-pathway oligomers for cryo-EM**. For the preparation of the oligomers$_1$ for cryo-EM, 200 μM lyophilized monomeric α-Syn was dissolved in sodium phosphate buffer, pH 7.4, and incubated for 20–24 h at 37 °C with constant agitation in the presence of 8 μM heme (preincubation). For the preparation of the untreated on-pathway oligomers, 200 μM monomeric α-Syn dissolved in sodium phosphate buffer, was incubated at 37 °C without agitation for 20–24 h, after which it was ultra-centrifuged at 45,000 rpm for 2 h and the supernatant was collected carefully while fibrils accumulated in the pellet were discarded. The excess monomeric species was removed by multiple filtrations using a 100 kDa cutoff centricon (Amicon, Merck Millipore, Burlington, MA, USA) to enrich the population of the oligomeric species. For the preparation of oligomers$_2$, 200 μM prefibrillar and fibrillar aggregates (that had been previously incubated for 48 h at 37 °C with constant agitation in the absence of heme) were further incubated for ~2 h at 37 °C with constant agitation in presence of 8 μM heme.

**Preparation of preformed fibril seeds**. For the intracellular seeding experiments as well as Thioflavin T seeding assay, α-Syn seeds were produced by incubating 200 μM monomeric α-Syn protein for 96 h under aggregation-inducing conditions, at 37 °C under constant agitation, in the absence or presence of 8 μM heme, after which the protein samples were sonicated in a bath sonicator for 15 min. For the Thioflavin T assay, 2 μM of the untreated or heme-treated sonicated fibril seeds were added to 200 μM fresh α-Syn monomers, and further incubated for a period of ~72 h.

**Thioflavin T assay**. The formation of cross-β structure during the aggregation of α-Syn was measured by the addition of 20 μM Thioflavin T (ThT) fluorescent probe dissolved in buffer to 2 μM protein aliquots collected from the incubation mixture at different timepoints[55]. Changes in the emission fluorescence spectra recorded between 450 and 520 nm with the excitation wavelength set at 440 nm were monitored using a Photon Technology International fluorescence spectrometer. For the control experiment on the effect of heme on ThT fluorescence, increasing concentrations of heme were added to ThT dissolved in 20 mM sodium phosphate buffer, pH 7.5, and the resulting fluorescence spectra were recorded.

**Negative-stain electron microscopy**. Five microliters aliquots taken from aggregation reactions were adsorbed onto 300 mesh carbon-coated copper grids (Agar Scientific, Stansted Essex, UK) and negative stained with 5 μl 2% (w/v)

uranyl acetate. Images were obtained at various magnifications (1000–90,000X) using a JEM-2100F 200 kV FE (Field Emission) transmission electron microscope.

**Atomic force microscopy**. Five microliters aliquots from the aggregation reactions were adsorbed onto a freshly cleaved muscovite mica (Agar Scientific, Stansted Essex, UK), followed by mild washing with 100 μl MilliQ water. The adsorbed α-Syn (treated or untreated with heme) were imaged by Acoustic Alternative Current or AAC (tapping) mode using an Agilent Technologies Picoplus AFM 5500. The scan frequency was set at 1.5 Hz. A 9-μm scanner was used. The heights of the protein structures were measured from topographic images using the PicoView 1.20.2 software (Molecular Imaging Corporation, San Diego, CA, USA).

**Cell culture, α-Syn-EGFP transfection into SH-SY5Y neuroblastoma cells and treatment with untreated and heme-treated fibril seeds**. The neuroblastoma cell line SH-SY5Y was maintained in DMEM supplemented with 10% heat-inactivated fetal bovine serum (FBS), 110 mg/L sodium pyruvate, 4 mM l-gluta-mine, 100 units/ml penicillin, and 100 μg/ml streptomycin in humidified air containing 5% CO2 at 37 °C. The cells were transiently transfected with 2.5 μg wild-type α-Syn-EGFP construct using Lipofectamine LTX and Plus reagent (Invitrogen) as described in the manufacturer's protocol. Twenty four hours before the transfection, cells were seeded on 35 mm poly-D-lysine coated plates (MatTek Corporation, Ashland, MA, USA) and allowed to grow till they were ~60% confluent. For inducing the aggregation of the fusion protein, the cells were transduced using Lipofectamine with 2 μM fibril seeds dissolved in Opti-MEM for 24 h, after which they were washed twice with Dulbecco's phosphate buffered saline (DPBS) and subjected to confocal imaging. α-Syn fibril seeds were prepared from the recombinant protein subjected to 96 h of aggregation, after which it was sonicated in a water bath for 15 min.

**Confocal Microscopy**. These experiments were carried out using a Zeiss LSM 510 Meta confocal microscope equipped with a C-Apochromat 40 X (NA = 1.20, water immersion) objective and confocal images were acquired with 512 × 512 (pinhole aperture ~1 airy units). The α-Syn-EGFP protein was excited using an argon laser at 488 nm.

**Preparation of SUVs (small unilamellar vesicles)**. Monomeric and oligomeric α-Syn interact with acidic phospholipids, although the effects of oligomers on the dynamic properties of synthetic lipid vesicles also depend on the additional presence of neutral phospholipids[56,57]. SUVs (small unilamellar vesicles) were used as α-Syn oligomers show a strong binding affinity to the augmented curvature of SUVs compared to that of LUVs (large unilamellar vesicles) and GUVs (giant unilamellar vesicles)[56]. Calcein-loaded SUVs of the composition of 3: 7 POPC: DOPS were added to α-Syn at a protein: lipid ratio of 1:10. Within cells, the extracellular leaflet of the plasma membrane is mostly composed of neutral phosphatidylcholine (PC) lipids such as 1-palmitoyl-2-oleoyl-sn-glycero-3-phos-phocholine (POPC) or 1,2-dioleoyl-sn-glycero-3-phosphocholine (DOPC) and some sphingolipids. The intracellular leaflet is rich in negatively charged phos-phatidylserine (PS) lipids such as 1-palmitoyl-2-oleoyl-sn-glycero-3-phospho-L-serine (POPS) and 1,2-dioleoyl-sn-glycero-3-phospho-L-serine (DOPS)[58]. Thus, we used POPC and DOPS for our experiments in the physiologically-relevant ratio of 3:7 PC:PS[59]. Additionally, due to its presence within brain membranes, we chose PS instead of phosphatidylglycerol to increase the negative charge content of the vesicles. 50 mM calcein when encapsulated within the SUVs is self-quenched, and hence shows a basal fluorescence at 515 nm when excited at 490 nm. 1 μl Triton X-100 was used to determine 100% calcein release, and all results were normalized to this value. For vesicle formation, a 3:7 ratio of POPC: DOPS was dissolved in 1 ml chloroform, followed by evaporation of the solvent under a stream of $N_2$ gas. The resulting lipid film was hydrated in 20 mM sodium phosphate buffer, pH 7.4 containing 50 mM calcein dye. The lipid-calcein suspension was sonicated in a glass tube in the dark at 40% amplitude for 30 minutes with 30 second pulse on and 1 minute pulse off at room temperature until the sample was transparent yellow in colour. The SUVs were isolated from free the dye by dialysing them in the dark in 20 mM sodium phosphate buffer, pH 7.4. The SUVs had a hydrodynamic diameter of 35 ± 10 nm according to dynamic light scattering measurements. The hydro-dynamic radius estimation was done using Malvern particle size analyser (Model no. ZEN 3690 ZETASIZER NANO ZS 90).

**Cytotoxicity assays**. For the FITC-Annexin V/ Propidium Iodide early and late apoptosis analyses, cells were seeded in 6 wells plates at $1 \times 10^6$ cells/ well. Twenty four hours after seeding, the cells were subjected to untreated α-Syn monomeric, prefibrillar and fibrillar structures (Fig. 2D) to understand which structures were the most toxic. Additionally, cells were also treated with either oligomers$_1$ (pre-incubation; prepared by incubating 200 μM α-Syn monomers with 8 μM heme added from the beginning and then incubated for n hours, where $n$ = 24, 36, 48, 60, 72, and 96 h, Fig. 2E), or oligomers$_2$ (postincubation; prepared by aggregating 200 μM α-Syn aggregated for n hours followed by treatment with 8 μM heme, where $n$ = 24, 36, 48, 60, 72, and 96 h, Fig. 2F). For comparison, a sample with equivalent heme concentration was used as the control. The final concentration of the protein added to the cells was maintained at 10 μM. After addition of the

treatments, the cells were incubated for 24 h at 37 °C, 5% CO$_2$. The percentage of apoptotic cells was determined using the InvitrogenDead Cell Apoptosis Kit following instructions from the manufacturer. Apoptosis is a cellular process that entails a genetically programmed series of events leading to the death of a cell. During early apoptosis, the lipid phosphatidylserine (PS) is translocated to the outer side of the plasma membrane from the cytoplasmic side. FITC-conjugated Annexin V is a strong probe for the exposed PS and can thus be used for detecting early apoptosis in stressed cells[60].

For the determination of late apoptosis and/or necrosis as a result of oligomer treatment, propidium iodide (PI) was added to the treated cells at a concentration of 2 μg/mL.PI labels the cellular DNA in late apoptotic/ necrotic cells where the cell membrane has been totally compromised.

**FT-IR analysis**. FT-IR measurements of the protein samples were performed in D$_2$O-containing sodium phosphate buffer, pH 7.5 on a Bruker FT-IR TENSOR 27 spectrometer. For the experiments, in order to study the effect of the heme on the aggregation of α-Syn, 200 μM of the monomeric protein was incubated without or with 8 μM heme under aggregation-inducing conditions (37 °C, 180 rpm) for 20–24 h. Conversely, to study the disaggregation of mature fibrils due to post-incubation with heme, another set of reactions was set up in which 200 μM fibrillar structures (that had been incubated for 48 h without heme) were further incubated with 8 μM heme. For all the three conditions, the protein was passed through a SEC column and only the oligomeric fractions were pooled and subjected to FT-IR. The protein concentration for each reading was maintained at 100 μM. The deconvoluted FT-IR spectra of proteins in the Amide I region (1700–1600 cm$^{-1}$) is due to the C = O stretching vibrations of the peptide bonds. It is predominantly informative about the backbone conformation and the relative composition of secondary structure elements of the protein[36]. The buffer background was independently measured and subtracted from each protein spectrum before curve fitting of the Amide I region. Data presented are a result of the fitting of three independent FT-IR spectra for each protein sample. In order to estimate the relative fraction of β-sheet content in each protein sample, deconvolution analysis with Gaussian/Lorentzian curves for each spectrum recorded was performed. The different Gaussian distributions were consigned to contributions from either β-sheet secondary structure or turns, or random coil according to the position of their peaks[61]. The analysis of the FT-IR spectra allows the distinction between parallel and antiparallel β-sheet structures based on the analysis of the amide I (1700–1600 cm$^{-1}$) region[62]. In antiparallel β-sheet structures, the amide I region displays two typical components: the major low frequency one has an average wavenumber located at ~1620–1638 cm$^{-1}$, whereas the minor high frequency component, 4–5-fold weaker than the major one, is characterized by an average wavenumber at 1695 cm$^{-1}$. For parallel β- sheet structures, the amide I region displays only the major component around 1620–1638 cm$^{-1}$. The oligomers and the fibrils reflect a high resemblance in the type of secondary structure organization except that they differ in the extent of β-sheet versus disordered content and the overall β-sheet arrangement (parallel for the fibrillar state and antiparallel for the oligomeric state). Turns are associated with various bands between 1660 and 1690 cm$^{-1}$, while unordered regions and loops are represented by bands around ~1642–46 cm$^{-1}$ and ~1656–64 cm$^{-1}$, respectively. Solvent subtraction, self-Fourier deconvolution of the Amide I region, determination of band position and curve fitting were performed using OriginPro 8.5 software (OriginLab Corp., Northampton, MA, USA).

**Heme-α-Syn binding**. In order to study the binding of heme to α-Syn, we incorporated a single cysteine mutation at the C-terminal of α-Syn to which a maleimide-containing dye could be attached. The binding assays of TMR-5-maleimide-tagged α-Syn-G132C or H50Q/G132C was investigated at 25 °C by fluorescence spectroscopy. The tagging of the protein using the dye was performed following the manufacturer's protocol. The measurements were performed on a Photon Technology International fluorescence spectrometer using a 1.0 cm path-length cell. Titrations were performed by adding increasing concentrations (0.1–4 μM) of heme dissolved in sodium phosphate buffer, pH 7.4, to a fixed amount (0.1 μM) of α-Syn-G132C or H50Q/G132C tagged with Tetramethylrhodamine-5-maleimide (ex/em 550 nm/ 574 nm in sodium phosphate buffer. The quenching efficiency was then evaluated by the Stern-Volmer quenching constant (K$_{SV}$), which is calculated from the following equation:[63]

$$F0/F = 1 + KSV[Q] \tag{1}$$

where F$_0$ and F are the emission intensities of α-Syn-TMR-5-maleimide in absence and presence of different concentrations of heme, and [Q] is the concentration of heme (quencher). A plot of F$_0$/F versus [Q] yields a slope equal to the Stern-Volmer quenching constant (K$_{SV}$). The relationship between the fluorescence intensity of the dye-tagged protein and the concentration of quencher was utilized to obtain the association constant (K$_a$) and the number of binding sites (n), both of which were calculated from the following equation:

$$\log[(F0 - F)/F] = \log Ka + n \log[Q] \tag{2}$$

where, K$_a$ refers to the association constant. The dissociation constant K$_d$ = 1/ K$_a$.

**Nuclear magnetic resonance**. One-dimensional proton NMR data was collected using the Topspin v3.5 software (Bruker Biospin GmbH, Switzerland) on the AVANCE III 700 MHz spectrometer, equipped with an RT probe. With a sweep width of 12 ppm, all the experiments were performed with a total of 64 scans each. 1D data was collected for 0.05 μM heme alone and upon subsequent titrations with WT α-synuclein and the mutant H50Q protein from equimolar concentrations to a molar ratio of 1:50. Trimethylsilylpropanoic acid (TSP) was used for the reference at 0 ppm. The experimental temperature was maintained at 283 K to eliminate any chance fibrillation during the experimental time frame. The data was further processed using the Topspin V4.0.6 software (Bruker Biospin GmbH, Switzerland). The individual peaks at the different chemical shift frequencies (in ppm) were picked for the calculation of their intensities. The data for the control (heme alone) were used to normalize the intensities for the peaks upon subsequent titrations and the ratio of peak intensities (I/I$_0$) in the presence (I) and absence of proteins (I$_0$) was calculated and plotted. The intensity ratios for the WT titrations were fit to following equation:

$$y = y_0 + \left( \frac{a*[\text{protein}]}{K_D + [\text{protein}]} \right) \tag{3}$$

where, y$_0$ is the intensity of free heme and a is a constant parameter defined as max (y)-min(y). For the new peaks that appeared in the fibrillar sample in the presence of heme, the last spectrum obtained at 44 h (t = 44 h) was used as a reference to observe the time-dependent signal intensity change from I (t = 0 h) to I$_0$ (t = 44 h). Similarly, the slight peak broadenings (I) observed in the presence of heme was also normalized with respect to the peak intensity at 0-h I$_0$ (t = 0 h). The intensity ratios (I/I$_0$) were then plotted together.

**Determination of peroxidase activity**. For the estimation of the peroxidase activity of α-Syn in the absence and presence of heme, 2 μM heme bound or unbound α-Syn (WT or H50Q, α-Syn/heme 25:1) was treated at 25 °C with 200 μM H$_2$O$_2$ and 10 mM guaiacol. The product formation kinetics was followed using a Shimadzu 1700 Pharmaspec UV-VIS spectrophotometer. The rate of decomposition of hydrogen peroxide (H$_2$O$_2$) by peroxidase using guaiacol as a hydrogen donor, was determined by measuring the rate of colour development spectrophotometrically at 470 nm using the extinction coefficient of 2.66 × 10$^4$ M$^{-1}$ cm$^{-1}$ [64].

**Labelling of α-Syn with Alexa488Maleimide, and the standard proteins BSA, HRP with 5-TAMRA, SE**. α-Syn does not contain any cysteine residues. We therefore introduced a cysteine mutation (G132C) at the C-terminal of the protein. The biophysical properties of the G132C mutant was found to be identical to that of the wild-type protein[65]. This cysteine mutant was then labeled with Alexa Fluor 488 C$_5$ Maleimide following the manufacturer's protocol. The reference proteins for molecular weight estimation were labelled using the amine-reactive 5-Carboxytetramethylrhodamine, succinimidyl ester, single isomer dye, using the manufacturer's protocol.

**Calculation of theoretical hydrodynamic radius (r$_{H\ Theo}$) using HullRad**. The HullRad software estimated the r$_G$ of the tetrameric model obtained from PDB 2N0A having a total number of 560 amino acid residues (140 residues/ monomer), to be 4.31 nm. we calculated the r$_H$ of the same to be 6.07 nm from its r$_G$ of 4.31 nm, using the formula:[66]

$$r_G^2 = \left( \frac{r_H^2}{2} \right) + \left( \frac{L^2}{12} \right) \tag{4}$$

where, L = length of the polymer, which in this case = 9 nm (from cryo-EM data).

**Fluorescence correlation spectroscopy**. FCS has been used extensively to study biomolecular interactions, aggregation, and protein conformational dynamics at single-molecule resolution[67]. For the experiments 20 nM labelled monomeric α-Syn, or 20 nM labelled α-Syn incubated with 200 μM unlabelled α-Syn and 8 μM heme from the beginning for ~24 h (oligomers$_1$) or 20 nM labelled α-Syn incubated with 200 μM unlabelled α-Syn for 48 h after which 8 μM heme was added (oligomers$_2$) were subjected to FCS analysis. FCS was performed using an ISS Alba FFS/FLIM Confocal Microscope (Champaign, IL, USA) with pulsed 488 nm diode and 532 nm fiber lasers focussed on a C-Apochromat 63 X water immersion objective. The emission intensity was collected using a pair of SPAD (Single Photon Avalanche Detector) detectors. The use of two detectors enabled us to determine single color cross correlation functions. In a simple diffusion experiment involving diffusing molecules (excluding the contributions of the triplet state), correlation functions G(τ) can be defined by the equation[68]

$$G(\tau) = 1 + \frac{1}{N} \cdot \left( \frac{1}{1 + \left( \frac{\tau}{\tau D} \right)} \right) \cdot \left( \frac{1}{\sqrt{1 + S^2 \left( \frac{\tau}{\tau D} \right)}} \right) \tag{5}$$

where τ$_D$ denotes the diffusion time of the diffusing molecule, N is the average number of molecules within the observation volume, and S is the structural parameter that defines the ratio between the radius and the height. The value of τ$_D$

obtained by fitting the correlation function is related to the diffusion coefficient (D) of a molecule by the following equation

$$\tau_D = \omega^2/4D \tag{6}$$

where $\omega$ is the size of the observation volume. Additionally, the value of the hydrodynamic radius ($r_H$) of the protein molecule/ complex/ aggregate can be obtained from D using the Stokes−Einstein formalism

$$D = kT/6\pi\eta r_H \tag{7}$$

where $\eta$ is the viscosity, T is the absolute temperature and k is the Boltzmann constant.

From the FCS experimental $\tau_D$ values obtained we calculated the $r_{H\,Exp}$ of the monomer to be 2.82 nm while for oligomer$_1$ and oligomer$_2$ it was 5.87 nm and 6.08 nm, respectively, which are in good agreement with the $r_{H\,Theo}$ value.

**SEC-MALS**. Samples were applied onto a Bio Rad ENrich SEC 70 column equilibrated with buffer containing 50 mm sodium phosphate, 150 mm NaCl, 0.05% w/v NaN3, pH 7.5, and eluted at 0.35 ml/min using a Shimadzu HPLC pump (Shimadzu, Kyoto, Japan). Absolute molecular weights were determined by static light scattering using a Wyatt Dawn 8 multiangle light scattering detector (Wyatt Technology Europe GmbH, Dernbach, Germany). The protein concentration data used to obtain molecular weights from light scattering data were derived from refractive index measurements (dRI detector, Wyatt Optilab, connected downstream of the LS detector). Sample injection and data collection using the ASTRA software were synchronized via the analog auto-inject signal; and the HPLC software was used to control the HPLC system. The standard value of dn/dc = 0.185 ml/g was used for the proteins.

**Native PAGE**. The fluorescently labeled α-Syn monomer, α-Syn oligomers$_1$ and oligomers$_2$ protein samples that were subjected to FCS were subsequently run on a 10% nondenaturing polyacrylamide gel. The wells were loaded with 20 μg labelled along with 20 μg unlabelled protein. Twenty micrograms of labelled horseradish peroxidase (HRP) and Bovine Serum Albumin (BSA) were used as protein standards.

**Cryo-electron microscopy**. Samples (~5 μl) were applied on glow-discharged lacey grids (Ted Pella, Redding, CA, USA), followed by blotting and vitrification of the grids using Vitrobot™(FEI, Hillsboro, OR, USA)[69]. Image data collection was performed on a Tecnai POLARA microscope (FEI, Hillsboro, OR, USA) equipped with a FEG (Field Emission Gun) operating at 300 kV. Images were collected with 4 × 4 K 'Eagle' charge-coupled device (CCD) camera (FEI, Hillsboro, OR, USA) at ~79,000X magnification (with defocus values ranging from ~1 to 5 μm), resulting in a pixel size of 1.89 Å at the object scale. All images were acquired using low-dose procedures with an estimated dose of ~20 electrons per Å$^2$ [70] Micrograph screening and particle picking were done separately using EMAN2[38] and SPIDER[39]. 60 micrographs for Sample 1 (α-Syn preincubated with heme for 24 h), 41 micrographs for Sample 2 (α-Syn late-incubated with heme after 48 h of aggregation), and 118 micrographs for Sample 3 (α-Syn incubated without heme for 24 h) were selected for particle picking. The initial models for the three different samples were developed using EMAN2 (by common line approach).

Validation of the 3D models was done in multiple ways (Supplementary Figs. 6 and 7, SI: schematic representation of the cryo-EM data processing). Particles formed following heme treatment appeared to be very small (~5–8 nm). We imaged three sets of data for each of the samples (each prepared and imaged on different dates) to confirm the sizes of the particles, particularly for oligomers preincubated with heme (sample 1), which showed distribution of particles of small sizes. The different views of the reference-free 2D class averages generated by different image processing softwares, e.g., EMAN2, Xmipp[71], RELION[72] as well as reference-based SPIDER were very similar. The initial models generated in EMAN2 were used as references for SPIDER auto picking after which 15770, 18626, and 11515 good particles were selected manually from auto-picked particles of Samples 1, 2, and 3, respectively. The 3D reconstruction was then performed following the standard SPIDER protocol for reference-based reconstruction[39] for the dataset of oligomers$_1$, oligomers$_2$ and the semi-annular oligomers (refer Supplementary Fig. 11, SI). The 2D re-projections of 3D maps were consistent with the 2D class averages. The overall resolution of three maps generated for samples 1, 2, and 3 were 15, 12.0, and 16.0 Å, respectively, using the FSC 0.5 cutoff criteria (10.4, 8.9, and 11.0 Å using the 0.143 cutoff criteria, Supplementary Fig. 7).

We also manually produced another starting model from the Greek-key structural motif (as a tetramer) by low pass filtering. The density model showed no distortion at the head and base junction. Final 3D map generated from dataset of sample 1 (oligomers$_1$) using this initial model showed similar the 'molecular mace' architecture with a distortion at the head and base junction. Details on the data collection and processing have been tabularized in Supplementary Table 2, SI.

Surface rendering, docking of crystal structures, segmentation and analyses of the 3D maps were performed in the program UCSF Chimera[73], Pymol[74], and VMD[75].

**Docking and molecular dynamics simulation**. We chose four chains of 2N0A (from residues 38–97, which form the well-structured region of the model) for molecular docking with hemin chloride (Chem spider ID: 16739951) and restricted the search space to the close vicinity of His50 residues by using AutoDockTools and AutoDock Vina[76]. For the initial docking studies smaller grid size covering the four histidine residues were considered to restrict the search locally around the histidine residues. The residues within the grid box were kept flexible for better optimization. For the concluding analyses bigger grid sizes with higher exhaustiveness were used for global search. We then selected the model with the highest rank and affinity as a starting structure for molecular dynamics simulation in Gromacs v 5.1.1[77] with the GROMOS96 54a7 force field, spc 216 water model and NaCl used for solvation and neutralization of net charges in the protein and ligand in a dodecahedron box. The steepest descent algorithm with a maximum of twenty thousand steps was applied for energy minimization of the system. Leap-frog integrator was utilized for the molecular dynamics run with constant temperature and pressure of 308 K and 1 bar, respectively, after proper equilibration of the system for 1 nsec. Simulation production run was carried out for 300 nsec. After the initial increase, trajectory with constant RMSD from the energy-minimized structure was considered for the analysis of stable conformations. UCSF Chimera was used for making structure visualization and superimposition[78].

**Statistics and reproducibility**. Data were generated from independent experiments, with datasets measured on different days; error bars represent standard error of the mean. Samples sizes are indicated in the methods section, figure legends, and are as follows: Thioflavin T fluorescence experiments (independent datasets = 8), FACS and calcein release (datasets=6, replicates for each condition subjected to cells/ SUVs = 3), FT-IR (independent datasets = 4), cryo-EM data acquisition (data for each of the following three datasets: oligomer$_1$, oligomer$_2$ and untreated oligomer were collected on three independent occasions), steady-state quenching and binding experiments (datasets = 6 for WT α-Syn and 3 for H50Q), FCS and SEC-MALS (datasets = 3 each), peroxidase assay (independent experiments performed on three occasions).

**Reporting summary**. Further information on research design is available in the Nature Research Reporting Summary linked to this article.

## Data availability

Deposition of the cryo-EM density maps in the Electron Microscopy Data Bank (EMDB) has been done. The accession numbers of the density maps of the heme-treated α-Syn oligomer$_1$, oligomer$_2$, and the oligomer formed without heme treatment are EMD-31004, EMD-31024, and EMD-31023, respectively. Source data for all graphs has been made available in Supplementary Data 1. Any remaining information can be obtained from the corresponding author upon reasonable request.

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

## Acknowledgements

This work was supported by Council of Scientific and Industrial Research (CSIR), Govt. of India's Network project 'UNSEEN' (BSC0113), CSIR's 02(0292)/17/EMR-II, SERB, Department of Science and Technology, Govt. of India-sponsored project CRG/2019/001788, as well as Department of Biotechnology-sponsored projects BT/PR21226/MED/122/41/2016 and BT/PR24905/NER/95/901/2017. We are grateful to the Director, CSIR-Indian Institute of Chemical Biology for his continued support. We appreciate the help from the Central Instrumentation Facilities (CIF) and the technical persons attached to it at CSIR-IICB. We sincerely thank Mr. Chiranjit Biswas for the cryo-EM data collection of the protein-heme complexes. We acknowledge Mr Thangamuniyandi Muruganandan, CIF, CSIR-IICB and Mr Supriya Chakraborty, CSS, Indian Association for the Cultivation of Science, for their assistance with the AFM and TEM imaging. R.C., S.D., P.S., S.S.P., and D.B. acknowledge CSIR and UGC, Govt. of India, for their senior research fellowship.

## Author contributions

K.C. conceived the project. K.C., R.C., S.S.P., and P.S designed the experiments in consultation with J.S. for the cryo-EM section. R.C. carried out the biochemical (TEM, AFM, fluorescence measurements, electrophoresis), biophysical (including FCS, SEC-MALS, FT-IR) and all cell-based experiments. S.D. performed elaborate 3D cryo-EM image processing and validation of the maps. P.S. performed initial biochemical experiments and preliminary image processing of the cryo-EM data with assistance from S.S.P. S.S.P. performed the molecular dynamics simulations. D.B. and A.B. performed the NMR experiments and analyses. K.C., J.S., R.C., and S.D. analysed the data. R.C. wrote the paper with inputs from K.C. and J.S. R.C. and S.D. contributed equally to the paper.

## Competing interests

K.C. is an Editorial Board Member for *Communications Biology*, but was not involved in the editorial review of, nor the decision to publish this article. All other authors declare no competing interests.
