## [Peer Review File · Communications Biology]

Reviewers' comments:

Reviewer #1 (Remarks to the Author):

The work entitled: "Conformational distortion in a fibril-forming oligomer arrests alpha-Synuclein fibrillation and minimizes its toxic effects", led by professor K. Chattopadhyay, shows how a heme molecule is capable of inhibiting α -synuclein fibril formation (a very dangerous component) and stabilize this protein in its oligomeric forms (less toxic protein aggregates). Furthermore, this molecule can reverse the fibrillation process. This process is undoubtedly interesting and well-known since small molecules may exert their therapeutic action by interfering with the pathogenic α -synuclein aggregation pathways, directing them to the formation of innocuous oligomers or fibrils, or by disassembling existing toxic aggregates, turning them into non-toxic forms. Derivatives of the heme group have been used and characterized: Hayden et al. *Biochemistry* 2015, 54, 30, 4599-4610. But they have not been as deeply studied as in this work, giving a set of interesting hypotheses for the development of drugs against the disease. I have read this interesting work in detail and care, but I did not find any notable point for criticism. A minor aspect is the quality of the figures that in some points are blurred, but I am convinced that in the final version it will be fixed. I suggest to the authors a use of molecular dynamics simulations since the way of fitting the PDBs in the forms provided by Cryo-EM are too coarse and could be a source of improvement in future works.

Reviewer #2 (Remarks to the Author):

Chakraborty et al. investigate oligomer formation of alpha-synuclein (asyn) in the presence of heme by ThT aggregation kinetics, EM and AFM imaging, cell culture studies, FTIR spectroscopy, and cryo-EM. The effect of heme on asyn aggregation including the characterization of resulting oligomers and their effects on membrane permeabilization and cytotoxicity has been reported before and these studies are not cited and discussed. Furthermore, there are some technical issues in data interpretation and the structural interpretation is highly speculative. What remains as new finding is that His-50 is required for the heme effects, as concluded from the absence of oligomer formation in the H50Q mutant. This seems to be not too surprising considering that His-50 is the only His, that Cys residues are absent, and that His-50 has been shown to be critical for certain metal interactions. Furthermore, the relevance of heme interaction of asyn for synucleinopathies is unclear.

- 1) The effect of heme on asyn aggregation, fibril dissociation, and characterization of the oligomers including their effects on membrane permeabilization and cytotoxicity has previously been reported by Hayden et al. "Heme stabilization of α -Synuclein oligomers during amyloid fibril formation" *Biochemistry* 2015 and Liu et al. "Hemin as a generic and potent protein misfolding inhibitor" 2014 BBRC. These papers are not cited and discussed.
- 2) The different techniques applied for oligomer characterization do not provide a consistent picture.
 - 2a) Since the initial characterization of asyn as an intrinsically disordered protein (Weinreb et al. 1996 *Biochemistry*) it has been realized that monomeric asyn elutes from SEC with an apparent MW of ~60 kD. Unfortunately, this unusual elution property due to the large hydrodynamic radius of the disordered protein has sometimes been mistakenly taken as evidence for a tetrameric asyn oligomer. In this manuscript, an apparent MW of 66.5 kD is taken as evidence for a tetramer, which however separates quite nicely from the apparent monomer. This elution profile of the monomer is unusual.
 - 2b) Even when considering that objects in AFM appear dilated due to tip-object convolution, the width of the oligomers by AFM (up to 50 nm) is too large for a tetramer and too large to correspond to the 1-3.5 nm wide objects seen in EM. These oligomers are already visible in the AFM image of monomeric asyn, questioning if the AFM objects are really heme-induced.
- 3) I.161 'Indicating inhibition of the primary nucleation-elongation micro-events'. This experiment does not allow to conclude that primary nucleation is inhibited, because primary nucleation is not

monitored in isolation but requires the fibril elongation reaction to become visible in ThT fluorescence. Inhibition of fibril elongation would be an alternative explanation. Another question is if ThT binding and fluorescence to asyn fibrils is modulated in the presence of heme. This should be checked.

4) I.184 Usage the term 'secondary nucleation' is incorrect here. This assay is not testing for secondary nucleation. In the figure legend it is correctly described as a disaggregation assay. The conclusion 'results demonstrate that heme can arrest the secondary micro-events of fibrillation' can therefore not be made.

5) There is an immediate decrease of ThT signal in the disaggregation assay. Is fibril disassembly that fast, or is that rather an effect of heme on ThT fluorescence? It should be possible to quantify the concentrations of asyn monomers and oligomers by SEC. Both for the fibril and oligomer formation and fibril disassembly reactions monomers and oligomers should be quantified by SEC, this would make these assays more informative.

6) Fig.2: The cell culture data is not very convincing. Punctate cellular inclusion are also visible in the sample w/o heme and in the control.

7) I.276 'The oligomers formed after 24 h of incubation in the absence of heme that showed augmented cytotoxicity (Figure 2D)' Oligomers were not isolated for Fig. 2D, so it is unclear why oligomers should be responsible for toxicity in this sample that was previously described to be fibrillar.

8) Fig.3a: The quantification of anti-parallel beta-structure content from the convoluted spectra is questionable. The total spectrum of Oligomer 2 looks very similar to that of the untreated sample, but the quantification of anti-parallel beta-structure gives a very different value, which is a consequence of how the deconvolution and band assignment was done.

9) The interpretation of the low-resolution cryo-EM data is highly speculative. Placement of 4 monomer units of the greek-key-fold into the density seems arbitrary and so is all further interpretation of the structure.

Minor points:

I.170 'corruptive protein templating or seeding' What is the meaning of 'corruptive' here?

I.171 'clumped oligomers (seeds)' Clumped oligomers sounds like oligomers of oligomers. This is not a common description for seeds. An amyloid fibril, for example, acts as a seed.

Reviewer #3 (Remarks to the Author):

The manuscript "Conformational distortion in a fibril forming oligomer arrests alpha-synuclein fibrillation and minimizes its toxic effects" by Ritobrata Chakraborty et al presented biophysical and structural insights into fibril-forming oligomers of alpha-synuclein stabilized by Heme. Heme appeared to interact with His50 residue and modulate the structure of oligomers and makes those to non-toxic oligomers.

Alpha-synuclein (AS) is an important protein linked with aggregation in the midbrain of a patient with PD. AS aggregates via multiple states where oligomers have postulated to be more toxic than monomer and oligomers. In this work, authors have reported that the addition of Heme at a sub-stoichiometric ratio to a monomeric or fibrillar state of AS stabilizes protein in the less-toxic conformation oligomeric form. Heme binds with His50 residue of the protein and locks protein into different conformation which leads to non-toxic oligomer formation. Further, the authors used CryoEM 3D reconstruction to suggest the mace-shaped structure of heme-stabilized oligomers. Using these structural details, authors proposed that upon binding, heme distort "Greek-key like" motif of the fibrillar structure to make less toxic oligomers.

This manuscript addresses an important issue in the field of aggregation and structure of transient oligomers. Paper is written nicely and the flow of the paper is logical. However, there are several shortcomings that must be addressed before consideration for publications. Here, I am summarizing in a point-wise manner:

1. Figures in the manuscript are of extremely poor quality. I wonder how can one submit such pathetic quality figures in the manuscript.
2. It is known that H50Q aggregates faster than WT. Any perturbation at His50 position is bound to have differential aggregation kinetics. Same authors have studied the interaction of Heme with AS. I am unable to appreciate the novelty of this work other than the CryoEM structural analysis.
3. How significant is it to report % of parallel and antiparallel beta strands using FTIR?
4. At many places, there are typographical errors, especially while writing temperature.
5. What are scale bars in Figure 1b, 1C etc.? Can TEM picture be taken a more zoom state? Oligomers may be clearly visible at higher magnification.
6. Why this composition of SUV (3:7 POPC DOPS) was chosen) any logic?
7. Authors have mentioned that "Heme converts antiparallel beta-sheet oligomers to parallel oligomers" however, at other places, it was mentioned that heme arrests the synuclein at the oligomer state and does not allow fibril formation. It looks contradictory, Is not it?
8. In figure 5, the H50Q in presence of heme should exhibit a profile similar to the only heme profile. Heme will not bind to H50Q and should carry out basal peroxidase activity as shown by only heme. However, authors have written, "Had no peroxidase activity in presence of heme either in figure 5B". Why?
9. In the method section authors mentioned about the G132C mutant, where is that used?
10. The conclusion must be written explicitly about this work. What is the outcome of the current work?

In summary, this manuscript attempted to present an important work however it lacks in the presentation. It must improve its scholarly presentation and address the above-mentioned points.

Response to the Reviewers' comments:

We sincerely thank all the reviewers for their thoughtful and supportive comments. We address the specific queries raised by the reviewers in this document and elaborate on the corresponding changes highlighted in the marked-up copy of the manuscript, which also contains figures placed next to their corresponding description. The final high-resolution figures have been provided as separate TIFF files in high resolution.

Reviewer #1 (Remarks to the Author):

The work entitled: "Conformational distortion in a fibril-forming oligomer arrests alpha-Synuclein fibrillation and minimizes its toxic effects", led by professor K. Chattopadhyay, shows how a heme molecule is capable of inhibiting a-synuclein fibril formation (a very dangerous component) and stabilize this protein in its oligomeric forms (less toxic protein aggregates). Furthermore, this molecule can reverse the fibrillation process. This process is undoubtedly interesting and well-known since small molecules may exert their therapeutic action by interfering with the pathogenic a-synuclein aggregation pathways, directing them to the formation of innocuous oligomers or fibrils, or by disassembling existing toxic aggregates, turning them into non-toxic forms. Derivatives of the heme group have been used and characterized: Hayden et al. *Biochemistry* 2015, 54, 30, 4599-4610. But they have not been as deeply studied as in this work, giving a set of interesting hypotheses for the development of drugs against the disease. I have read this interesting work in detail and care, but I did not find any notable point for criticism.

We are grateful for this positive assessment of our work.

A minor aspect is the quality of the figures that in some points are blurred, but I am convinced that in the final version it will be fixed.

We apologise for the poor quality of some of the figures, and have updated the figures with high resolution images as individual TIFF files.

I suggest to the authors a use of molecular dynamics simulations since the way of fitting the PDBs in the forms provided by Cryo-EM are too coarse and could be a source of improvement in future works.

We thank the reviewer for this excellent suggestion. The fitted model of our low resolution cryo-EM density map with four chains of the α -Syn PDB 2N0A¹ structure (Figure 6, main text) suggested that the reported Greek-key topology in 2N0A is required to be twisted for a reasonably good fit. We postulate that this twist in the Greek-key topology is crucial for heme-mediated arrest of intermediate oligomers as well as for destabilisation of the higher-order structures formed during α -Syn. We performed a molecular docking study followed by molecular dynamic simulations on a tetrameric model of the published PDB 2N0A structure to observe any changes caused by heme, as suggested by the reviewer. Indeed, a distortion in the tetrameric Greek-key architecture is observed in the final structure of heme-bound tetramer derived after simulation.

The result of our simulation studies is represented here in Figure R1 (also Figure S13, Supporting Information), where (A) shows the well-equilibrated and stable end-simulation structure of the tetramer with the heme molecule ligated at His 50. Figure (B) shows the superimposition of the end-simulation structure onto our cryo-EM fitted model from Figure 7, main text.

Figure R1: A well-equilibrated and stable end-simulation structure (R1A) shows the presence of the twist that matches with the cryo-EM fitted model of 2N0A (R1B).

Reviewer #2 (Remarks to the Author):

Chakraborty et al. investigate oligomer formation of alpha-synuclein (asyn) in the presence of heme by ThT aggregation kinetics, EM and AFM imaging, cell culture studies, FTIR spectroscopy, and cryo-EM. The effect of heme on asyn aggregation including the characterization of resulting oligomers and their effects on membrane permeabilization and cytotoxicity has been reported before and these studies are not cited and discussed. Furthermore, there are some technical issues in data interpretation and the structural interpretation is highly speculative. What remains as new finding is that His-50 is required for the heme effects, as concluded from the absence of oligomer formation in the H50Q mutant. This seems to be not too surprising considering that His-50 is the only His, that Cys residues are absent, and that His-50 has been shown to be critical for certain metal interactions. Furthermore, the relevance of heme interaction of asyn for synucleinopathies is unclear.

Our response: We thank the reviewer for analysing our manuscript. We have added few relevant citations, which were missing in the previous version (please refer to the references #18, 19). We have also modified the Introduction part of the manuscript to discuss the novelty and relevance of the present study (page 3, line 23 onward). Briefly, although the structural biology of amyloid fibrils-the end point of the aggregation of several neurodegenerative diseases causing proteins have been studied extensively, not much is done on the early intermediate species (or oligomers), which may have major contribution towards disease toxicity. To be specific, more than fifty 3-dimensional structures are now available for amyloid fibrils of different proteins², while there is only one low resolution structure available for the toxic oligomer³. This lack of structural understanding of the early oligomers- and the basis of their toxicity- limits any drug development efforts towards these species.

It is generally believed that the heterogeneity of the aggregation landscape and the interconversion dynamics of the early intermediate species are some of the predominating factors for the lack of their structural elucidation. In this paper, we have shown for the first time that the binding of heme to a His50 residue in the inter-protofilament interface of α -Syn, minimizes the heterogeneity significantly, enabling us to determine the low-resolution structure of a non-toxic intermediate. This paper also identifies a distortion in the oligomer structure-which heme binding induces-which may modulate its toxicity. Since His50Gln is a late onset PD mutant, targeting this residue by heme (or heme like molecule) to modulate toxicity may offer a novel approach towards small molecules development. We have discussed this at the page 15 of the modified manuscript.

In addition, the physiological relevance of heme- α -Syn system come from other literature data. Several heme-proteins have been reported to possess protective functions in neurodegenerative diseases. Cytochrome

c is present in abundance within α -Syn Lewy body neurites where the two proteins form an anti-apoptotic, peroxidase activity-positive covalently-bonded hetero-oligomer^{4,5}. Ferric cytochrome c has also been shown to inhibit α -Syn aggregation⁵. Under physiological conditions within the human brain and peripheral RBCs, the heme cofactor-containing neuronal haemoglobin scavenges α -Syn, leading to a reduction of PD-induced mitochondrial damage and apoptosis⁶. Additionally, over-expression of the heme-protein neuroglobin inside neuronal cells, reduces cytoplasmic α -Syn inclusions and associated mitochondrial damage⁷. We have discussed this at the page 3 of the modified manuscript.

1) The effect of heme on asyn aggregation, fibril dissociation, and characterization of the oligomers including their effects on membrane permeabilization and cytotoxicity has previously been reported by Hayden et al. “Heme stabilization of α -Synuclein oligomers during amyloid fibril formation” Biochemistry 2015 and Liu et al. “Hemin as a generic and potent protein misfolding inhibitor” 2014 BBRC. These papers are not cited and discussed.

Our response: We apologize. The references mentioned by the reviewer have been incorporated in our manuscript on page 3, line 40.

2) The different techniques applied for oligomer characterization do not provide a consistent picture.

2a) Since the initial characterization of asyn as an intrinsically disordered protein (Weinreb et al. 1996 Biochemistry) it has been realized that monomeric asyn elutes from SEC with an apparent MW of ~60 kD. Unfortunately, this unusual elution property due to the large hydrodynamic radius of the disordered protein has sometimes been mistakenly taken as evidence for a tetrameric asyn oligomer. In this manuscript, an apparent MW of 66.5 kD is taken as evidence for a tetramer, which however separates quite nicely from the apparent monomer. This elution profile of the monomer is unusual.

Our response: We agree with the reviewer that the elution profile of an intrinsically disordered protein like α -Syn will be different from what is expected for a well-folded globular protein. Considering the difficulty in determining MW of a non-spherical molecule using SEC, we used native gel electrophoresis (Figure R2, below, also Figure S8, SI) instead, to gain insight into the apparent MW of the heme-treated oligomers. We found that complimentary to established native PAGE analyses^{3,8,9} the α -Syn monomer migrated with a MW of ~15 kDa. The oligomers₁ and oligomers₂ migrate with a MW of ~ 60 kDa (when compared to the migration patterns of two reference proteins: Bovine Serum Albumin, 66.4 kDa and Horseradish peroxidase, 44 kDa). This data suggested that the heme-treated oligomers were probably tetrameric (MW of α -Syn monomer = 14.5 kDa). To support this claim, we estimated the theoretical hydrodynamic radius ($r_{H, Theo}$) of a tetrameric model of α -Syn obtained from its PDB structure (ID 2N0A) using the HullRad algorithm¹⁰. We then compared it with the experimental $r_{H, Exp}$ of the heme-treated oligomers calculated using fluorescence correlation spectroscopy or FCS (Figure R3, below, and also Figure S9, SI). We found that the theoretical $r_{H, Theo}$ value was in excellent agreement with the experimental $r_{H, Exp}$. This proved that the heme-induced oligomers were indeed tetrameric (Refer page 12, line 13 onward).

Figure R2: A 10 % Native PAGE of AlexaFluor488-tagged α -Syn monomer (MW ~15 kDa, first lane from left), 5-TAMRA, SE-tagged horseradish peroxidase (HRP, MW ~44 kDa, lane 2), AlexaFluor488-tagged α -Syn oligomers₁ (lane 3), AlexaFluor488-tagged α -Syn oligomers₂ (lane 4), 5-TAMRA, SE-tagged Bovine Serum Albumin (BSA, 66.4 kDa, lane 5). The monomer migrated at ~15 kDa as has been reported before^{3,8,9} while the oligomers₁ and oligomers₂ migrated at ~60 kDa. These fluorescently labeled proteins were also subjected to FCS to correlate the FCS r_H values with their observed migration pattern.

Figure R3: (A) FCS correlation curves of α -Syn monomer (open red circles) plotted with the heme-treated oligomers₂ (black circles) indicate an increase in the diffusion time (and hence the $r_{H \text{ Exp}}$) of the heme-treated oligomers. The data were fit to a single-component diffusion model (solid lines). (B) Plot of the \log_e of the hydrodynamic radius (r_H) versus the \log_e of the number of residues in the polypeptide chain. The line fitted to these data for the native folded proteins (black) has a slope of 0.29 ± 0.02 and a y-axis intercept of 1.56 ± 0.1 , while that fitted to the chemically denatured protein (green) data has a slope of 0.57 ± 0.02 and a y-axis intercept of 0.79 ± 0.07 . Literature data have been used for the folded and chemically denatured proteins¹¹

while we employed FCS to calculate the r_H of the pre-treated oligomers (oligomers₁), post-treated oligomers (oligomers₂). Both oligomers have a $r_{H, FCS}$ that lies between that of natively-folded and unfolded proteins.

2b) Even when considering that objects in AFM appear dilated due to tip-object convolution, the width of the oligomers by AFM (up to 50 nm) is too large for a tetramer and too large to correspond to the 1-3.5 nm wide objects seen in EM. These oligomers are already visible in the AFM image of monomeric asyn, questioning if the AFM objects are really heme-induced.

Our reply: This is a valuable comment and we thank the reviewer for bringing this up. An AFM image is a convolution of the shape of the probe and the shape of the sample. In AFM, the lateral resolution is low (~30nm) due to this convolution, while the vertical resolution can be up to 0.1 nm. This makes protruding features to appear wide, while holes appear smaller (both narrower and often less deep, too). Less sharp probes can enhance this effect. Thus, when the dimensions of an object are measured on AFM using the semi-contact method in air, an inflated estimate of the object width and an underestimated height of the object are obtained. Therefore, prior to the analysis of the object height using AFM, the tip of the cantilever is calibrated¹². But the measurements taking into account the size of the tip of the cantilever are not always carried out, which introduces errors in the parameters of the measured objects (Figure R4, below)

Figure R4: Dilation of lateral particle dimension during AFM imaging: The finite size of the scanning tip, shown here as a parabola, increases the apparent width (W_{app}) of the object well beyond its actual width (W). For an ellipse, W can be derived from the measured height H and W_{app} of the object and the tip radius R_t ¹².

Therefore, due to the aforementioned limitation, the lateral width of the object appears larger than its actual diameter. Hence, the height of the particles (in this case oligomers and fibrils) are usually reported^{13,14} instead of the width (refer AFM height histograms, Figure 1 B, C, F, main text) and have found that they match very well with the established reports on the AFM height as well as TEM diameter of the α -Syn aggregates.

3) 1.161 'Indicating inhibition of the primary nucleation-elongation micro-events'. This experiment does not allow to conclude that primary nucleation is inhibited, because primary nucleation is not monitored in isolation but requires the fibril elongation reaction to become visible in ThT fluorescence. Inhibition of fibril elongation would be an alternative explanation. Another question is if ThT binding and fluorescence to asyn fibrils is modulated in the presence of heme. This should be checked.

Our reply: We agree with the reviewer that by observing fibril elongation (which causes the ThT fluorescence to increase sharply upon the development of cross β structure) one cannot identify the primary

nucleation pathways in isolation. We have therefore, rewritten the section on page 5 line 13 onward. We have also checked the effect of heme on ThT fluorescence intensity alone (Figure S2, Supporting Information, also Figure R5, below) where we measured ThT fluorescence in presence of increasing concentrations of heme and found that the ThT intensity was unperturbed at the heme concentrations used in this study.

4) I.184 Usage the term ‘secondary nucleation’ is incorrect here. This assay is not testing for secondary nucleation. In the figure legend it is correctly described as a disaggregation assay. The conclusion ‘results demonstrate that heme can arrest the secondary micro-events of fibrillation’ can therefore not be made.

Our reply: The reviewer is correct in pointing out that individual micro-events that occur within the aggregation process cannot be demarcated by observing a change in ThT fluorescence alone. Indeed, chemical kinetic assays are now used to elucidate the micro-events that underlie the stages of an aggregation process¹⁵. We have therefore rewritten the section on page 5 line 41 onward.

5) There is an immediate decrease of ThT signal in the disaggregation assay. Is fibril disassembly that fast, or is that rather an effect of heme on ThT fluorescence? It should be possible to quantify the concentrations of asyn monomers and oligomers by SEC. Both for the fibril and oligomer formation and fibril disassembly reactions monomers and oligomers should be quantified by SEC, this would make these assays more informative.

Our reply: We have carried out ThT fluorescence experiments in the presence of heme (Figure R5, below and also Figure S2, SI) and have not observed any significant change in fluorescence intensity of ThT in the presence of the heme concentrations used in the present study. This rules out the possibility of the heme-induced effect on ThT fluorescence. Our AFM and TEM data also show that heme addition disaggregates α -Syn, thereby supporting the ThT results.

Figure R5: The concentration of heme (hemin chloride) used in our studies (which is 8 μ M) does not have a quenching effect on the fluorescence of 20 μ M Thioflavin T. Both hemin chloride and ThT were dissolved in sodium phosphate buffer, pH 7.5.

Due to the extended structure of the heme-treated oligomer, its characterization using SEC was found to be difficult (discussed in detail in our answer to the reviewer's question number 2a) as a result of which we have omitted it from the manuscript. Although we had planned elaborate experiments and analyses for the quantification of concentrations of the end products of α -Syn disaggregation upon heme addition, we were not able to complete them as our laboratory has been shut down due to the nationwide COVID-19 pandemic lockdown.

6) **Fig.2: The cell culture data is not very convincing. Punctate cellular inclusions are also visible in the sample w/o heme and in the control.**

Our reply: We have clarified this concern of the reviewer by providing magnified versions of the confocal images. The images pertaining to the α -Syn-GFP transfected cells to which heme-treated fibril seeds were added (Figure 3B) as well as the control, to which no seeds were added (Figure 3C), show that some amount of diffused fluorescence is associated with the cell periphery (plasma membrane). However, no fluorescent dots are noticed on the membrane. This is a result of the binding of monomeric α -Syn-GFP onto the membrane as has been observed before¹⁶. We have magnified the confocal images for better understanding and placed them below the original unmagnified images. (Figure R6, below and Figure 3A-C)

Figure R6: **Heme minimizes the *in-cell* seeding, synthetic SUV permeabilization and cytotoxicity of various aggregated species of α -Syn.** (A-C) α -Syn-EGFP-transfected SH-SY5Y cells were transfected and incubated for 24 h with 1 μ M fibril seeds prepared by sonicating pre-formed α -Syn fibrils that had been incubated under aggregation-inducing conditions (A) without or (B) with heme for 96 h (α -Syn/heme = 25:1). Punctate green structures (white arrows) within the cytoplasm in (A, bottom inset) denote seeded α -Syn aggregates. (C) Cells transfected with α -Syn-EGFP but not transfected with seeds (vehicle) do not convert into an inclusion-positive state. The scale bars denote 10 μ m. (D-F) The effect of α -Syn structural conformers formed after various

periods of aggregation (D) in the absence of heme, or (E) in presence of heme: where heme was added from the beginning of aggregation period (pre-incubation), or (F) when heme was added after various periods of aggregation had occurred (post-incubation), on calcein-loaded liposomal SUV permeation (black bars), early apoptosis (red bars) and late apoptosis/ necrosis (blue bars) as observed in SH-SY5Y neuroblastoma cells. Error bars are \pm SEM.

7) 1.276 ‘The oligomers formed after 24 h of incubation in the absence of heme that showed augmented cytotoxicity (Figure 2D)’ Oligomers were not isolated for Fig. 2D, so it is unclear why oligomers should be responsible for toxicity in this sample that was previously described to be fibrillar.

Our reply: We thank the reviewer for pointing out this detail. Indeed, for the experiment corresponding to Figure 2D, the oligomers were not isolated and the entire heterogeneous populations of oligomers, protofilaments (which are single stranded 5 nm wide filaments that later join linearly to form mature 10 nm fibrils) and other prefibrillar/ fibrillar structures formed after various ‘*n*’ hours of under constant agitation at 37°C were added to the cells after which the extent of their toxicity was measured. To clarify, when the incubation time period *n*= 24h and 36h, the aggregates are mostly oligomeric with a few short protofilaments, while at *n*= 48 h and 60 h, the populations of aggregates are more heterogeneous consisting of oligomers, protofilaments and short fibrils (protofibrils), while at *n*= 72 h and 96 h, the aggregates are mostly fibrillar with a diameter of ~8.5-10 nm. From the data presented in Figure 2D, it is clear that the mixed heterogeneous assemblage of aggregates formed between 36 and 60 h show maximum toxicity to both synthetic membranes and cells. We have made the necessary changes in the manuscript on page 18, line 40 onward.

8) Fig.4a: The quantification of anti-parallel beta-structure content from the convoluted spectra is questionable. The total spectrum of Oligomer 2 looks very similar to that of the untreated sample, but the quantification of anti-parallel beta-structure gives a very different value, which is a consequence of how the deconvolution and band assignment was done.

Our reply: To address reviewer’s concerns, we have discussed in the modified version the fitting strategy of the FT-IR data. For most cases, the band positions of FT-IR were determined also by the second derivative spectra, which has enhanced the separation of the overlapping peaks. The double derivative spectra of oligomers₂ and the untreated samples are now provided as Supporting Information, which show clearly the difference (Figure R7, below and Figure S5).

Figure R7: FTIR second derivative spectra of the α -Syn oligomers show the positions of overlapping peaks. In case of the untreated oligomers (in red), the presence of a marked band at 1695 cm^{-1} denotes the presence of antiparallel β sheet structure. This 1695 cm^{-1} band is noticeably absent in the spectra for the heme-treated oligomers₁ (in black) and oligomers₂ (in blue). Additionally, the band at $\sim 1685\text{ cm}^{-1}$ (also denotes antiparallel β sheet) is also reduced in case of the heme-treated oligomers. This is quantified by deconvoluting the IR spectra (refer Figure 3D, main text) where we find that the heme-treated oligomers show an obvious reduction in antiparallel β sheet content from 12% (untreated oligomers) to 2% (oligomers₁) and 5% (oligomers₂).

9) The interpretation of the low-resolution cryo-EM data is highly speculative. Placement of 4 monomer units of the greek-key-fold into the density seems arbitrary and so is all further interpretation of the structure.

Our reply: We understand the concern of the reviewer. Since the cryoEM structure is of low resolution, we have used several orthogonal methods to support our understanding of the data, which have been discussed below:

Linear shaped/elongated particles of heme treated oligomers have been found in negatively stained TEM images (Figure S3, SI and Figure R8, below). In agreement with that, cryo-EM images show distribution of particles having similar shape and size. 2D averages and 3D reconstructed maps also suggest the same. Thus, the formation of specific mace-like oligomers is confirmed. Additionally, the 3D reconstructed mace-like cryo-EM density maps match remarkably well in shape and size with the ‘Greek-key like motif’ identified in the PDB 2N0A structure.

Figure R8: TEM micrograph showing several elongated oligomers₂ (in green boxes), one of which have been further magnified to show its linear shape (inset).

Second, the molecular mass of the mace-like cryo-EM density map has been theoretically estimated as ~52 kDa from the size of the enclosed volume (at the threshold value shown in Figure S5), which appeared to be equivalent to ~4 α -Syn molecules (considering that the unstructured N- and C-terminals are only partially visible in the density maps).

Finally, we used fluorescence correlation spectroscopy (FCS), a convenient method to study biomolecular diffusion at single molecule resolution, to study extensively the hydrodynamics of the heme-induced oligomers. The hydrodynamic radius (r_H) of the monomeric protein was found to be larger than the expected value for a folded globule, and is close to the expected r_H of a chemically unfolded protein of similar molecular weight (Figure S9, SI and Figure R3A). From the FCS correlation curves (Figure R3B), we obtained the experimental values of the r_H ($r_{H,Exp}$) of the heme induced oligomers to be 5.9 nm and 6.1 nm respectively for oligomers₁ and oligomers₂. In addition, we used HullRad software to compute the value of the r_G of a tetrameric model obtained from the PDB 2N0A structure having a total number of 560 amino acid residues (140 residues/ monomer for a tetramer). Using the value of r_G (4.31 nm) determined using HullRad, we calculated the theoretical $r_{H,Theo}$ of the tetramer to be 6.07 nm using the formula¹⁷:

$$r_G^2 = \left(\frac{r_H^2}{2}\right) + \left(\frac{L^2}{12}\right)$$

where, L = length of the linear polymer, which in this case = 9 nm (cryo-EM data, Figure 5C and 5F, main text). The agreement between the values of $r_{H,Theo}$ (6.07 nm) and $r_{H,Exp}$ (5.9 nm and 6.1 nm for oligomers₁ and oligomers₂ respectively) strongly suggest that the heme-stabilized oligomers (oligomers₁ and oligomers₂) are indeed tetrameric.

Minor points:

l.170 ‘corruptive protein templating or seeding’ What is the meaning of ‘corruptive’ here?

Our reply: We thank the reviewer for their query. To make our point more comprehensible, we have replaced the word ‘corruptive’ with ‘toxic’ on page 5, line 21.

l.171 ‘clumped oligomers (seeds)’ Clumped oligomers sounds like oligomers of oligomers. This is not a common description for seeds. An amyloid fibril, for example, acts as a seed.

Our reply: We have made the necessary corrections as suggested by the reviewer on page 5, line 22.

Reviewer #3 (Remarks to the Author):

The manuscript “Conformational distortion in a fibril forming oligomer arrests alpha-synuclein fibrillation and minimizes its toxic effects” by Ritobrata Chakraborty et al presented biophysical and structural insights into fibril-forming oligomers of alpha-synuclein stabilized by Heme. Heme appeared to interact with His50 residue and modulate the structure of oligomers and makes those to non-toxic oligomers.

Alpha-synuclein (AS) is an important protein linked with aggregation in the midbrain of a patient with PD. AS aggregates via multiple states where oligomers have postulated to be more toxic than monomer and oligomers. In this work, authors have reported that the addition of Heme at a sub-stoichiometric ratio to a monomeric or fibrillar state of AS stabilizes protein in the less-toxic conformation oligomeric form. Heme binds with His50 residue of the protein and locks protein into different conformation which leads to non-toxic oligomer formation. Further, the authors used CryoEM 3D reconstruction to suggest the mace-shaped structure of heme-stabilized oligomers. Using these structural details, authors proposed that upon binding, heme distort “Greek-key like” motif of the fibrillar structure to make less toxic oligomers.

This manuscript addresses an important issue in the field of aggregation and structure of transient oligomers. Paper is written nicely and the flow of the paper is logical. However, there are several shortcomings that must be addressed before consideration for publications.

We appreciate the positive assessment of the significance of our manuscript. We are grateful for the critical evaluation of our manuscript. Below, we have addressed the specific comments.

1. Figures in the manuscript are of extremely poor quality. I wonder how can one submit such pathetic quality figures in the manuscript.

Our reply: We apologize for the poor quality of the figures. We have now improved their resolution and uploaded them as separate high-resolution TIFF files. Comparatively lower resolution files are also provided as embedded image in the marked copy of the manuscript.

2. It is known that H50Q aggregates faster than WT. Any perturbation at His50 position is bound to have differential aggregation kinetics. Same authors have studied the interaction of Heme with AS. I am unable to appreciate the novelty of this work other than the CryoEM structural analysis.

Our reply: We thank the reviewer for his concern and have clarified our justification for performing this study:

Since the His50 residue is crucial for the formation of a stable inter-protofilament bonding (known as the steric zipper interface) that joins two protofilaments, each of 5 nm diameter, into one mature 10 nm wide fibril, we aimed to destabilize this interface and cause fibril breakdown by using a physiologically-appropriate small molecule heme which specifically (and with a sub-micromolar K_D of 0.59 μM) binds to His50. Further physiological relevance of His50 comes from the following arguments:

Although the H50Q mutant protein has a faster aggregation kinetics compared to wild type (WT) α -Syn¹⁸, we show that blocking of the inter-protofilament His residue has an opposite effect that prevents fibrillation. We, therefore, propose a method of ‘blocking’ steric zipper residues using small molecules that may be harnessed to develop therapeutics. This novel mechanism of heme-induced defibrillation is two-fold: (i) prevention of the His50-mediated steric zipper bonding, and (ii) distortion/ twisting of the building ‘block’ or kernel (oligomers₁ and oligomers₂) of the protofilament (which we observed using cryo-EM and is now supported by additional data in the form of a MD simulation of α -Syn tetrameric kernel to which heme was docked, Figure S13, SI and Figure R1) which prevents further assemblage of these kernels into higher order structures.

3. How significant is it to report % of parallel and antiparallel beta strands using FTIR?

Our reply: Antiparallel β sheets (which are predominant within the toxic α -Syn oligomers) are more stable than parallel β sheets (present within the minimally-toxic fibrils) because of their more optimal hydrogen bonding pattern. Here, by reporting the percentage of parallel vs antiparallel β sheet content we describe a conformational switch from stable antiparallel β to less-stable parallel β sheet structure, which thereby lowers their cytotoxicity. The exact mechanism linking β sheet orientation and toxicity is yet not known and is being actively studied in our laboratory.

4. At many places, there are typographical errors, especially while writing temperature.

Our reply: We apologize. We worked hard to minimize these errors.

5. What are scale bars in Figure 1b, 1C etc.? Can TEM picture be taken a more zoom state? Oligomers may be clearly visible at higher magnificence.

Our reply: We are grateful for this comment. The scale bars in the further magnified TEM figures Figure 1B and 1C now represent 25 nm. We have included a zoomed in view of the heme-stabilized oligomers in Figure S3, SI, which we have reproduced in Figure R8 in this document as well.

6. Why this composition of SUV (3:7 POPC DOPS was chosen) any logic?

It is known that monomeric and some oligomeric conformers of α -Syn interact with acidic phospholipids, probably through the N-terminal lysine residues^{19,20} of the protein. Furthermore, the effects of the oligomers on synthetic lipid vesicles depend strongly on the relative proportion of acidic and neutral phospholipids^{21,22}. Therefore, we had initially experimented with α -Syn-induced membrane permeation using SUVs composed of various ratios of acidic and neutral phospholipids. Within cells, the extracellular leaflet of the plasma membrane is mostly composed of neutral phosphatidylcholine (PC) lipids such as 1-palmitoyl-2-oleoyl-sn-glycero-3-phosphocholine (POPC) or 1,2-dioleoyl-sn-glycero-3-phosphocholine (DOPC). The intracellular leaflet is rich in negatively charged phosphatidylserine (PS) lipids such as 1-palmitoyl-2-oleoyl-sn-glycero-3-phospho-L-serine (POPS) and 1,2-dioleoyl-sn-glycero-3-phospho-L-serine (DOPS)²³. Thus, we used PC and PS variants for our experiments, and in the ratio of 3:7 as this ratio is most relevant physiologically²⁴. Additionally, due of its presence within brain membranes, we chose PS instead of phosphatidylglycerol (PG) to increase the negative charge content of the vesicles.

We have added the aforementioned justification for the use of 3:7 POPC DOPS in the Experimental Procedures section on page 19, line 11 onward.

7. Authors have mentioned that “Heme coverts antiparallel beta-sheet oligomers to parallel oligomers” however, at other places, it was mentioned that heme arrests the synuclein at the oligomer state and does not allow fibril formation. It looks contradictory, Is not it?

Our reply: We thank the reviewer for this question. We would like to clarify our motive for performing the FT-IR spectroscopy for secondary structure analysis. Since oligomers are the most toxic species within the aggregation pathway of α -Syn, we aimed at understanding the structural foundations of their toxicity. Thus,

for the FTIR study we used only the oligomeric populations formed in the absence and presence of heme, and compared the results (refer lines 12-20, page 20, main text). We observed that the addition of heme arrests the protein in a non-toxic oligomeric state, which comprise predominantly of parallel β sheets.

8. In figure 5, the H50Q in presence of heme should exhibit a profile similar to the only heme profile. Heme will not bind to H50Q and should carry out basal peroxidase activity as shown by only heme. However, authors have written, “Had no peroxidase activity in presence of heme either in figure 5B”. Why?

Our reply: We thank the reviewer for pointing this error out. We redid the peroxidase assay of H50Q+heme to realize that it did elicit some amount of activity (page 11, line 30 onward). We have replaced the figure (Figure 5C) with the correct one.

9. In the method section authors mentioned about the G132C mutant, where is that used?

Our reply: We used the G132C mutant so that α -Syn could be tagged by a maleimide–fluorophore conjugate like tetramethylrhodamine-5-maleimide. Such dyes are highly specific for the thiol group of cysteine and the tagging reactions are rapid (10 min–2 h) in typical buffers and require moderate pH and temperatures. The G132C mutant has been used extensively for biophysical studies including in FCS studies and has the same properties as the WT protein. The details about the use of the G312C mutant are mentioned in the main text on page 12, line 17 onward as well as in the Experimental Section on page 21, line 21 onward.

10. The conclusion must be written explicitly about this work. What is the outcome of the current work?

Our reply: We have rewritten the Conclusion part as suggested by the reviewer on page 14 onward.

In summary, this manuscript attempted to present an important work however it lacks in the presentation. It must improve its scholarly presentation and address the above-mentioned points.

Our reply: We thank the reviewer for all the comments and suggestions. We have revised the figures and text in order to present the data in a better way. The modified version contains a revised conclusion and hopefully is a better written manuscript, which addresses all concerns of the reviewers.

References:

- 1 Tuttle, M. D. *et al.* Solid-state NMR structure of a pathogenic fibril of full-length human alpha-synuclein. *Nat Struct Mol Biol* **23**, 409-415, doi:10.1038/nsmb.3194 (2016).
- 2 Sawaya, M. R. *Amyloid Atlas*, <<https://people.mbi.ucla.edu/sawaya/amyloidatlas/>> (2020, September, 22).
- 3 Chen, S. W. *et al.* Structural characterization of toxic oligomers that are kinetically trapped during α -synuclein fibril formation. *Proc. Natl. Acad. Sci. U. S. A.* **112**, E1994 (2015).
- 4 Bayır, H. *et al.* Peroxidase Mechanism of Lipid-dependent Cross-linking of Synuclein with Cytochrome c: PROTECTION AGAINST APOPTOSIS VERSUS DELAYED OXIDATIVE STRESS IN PARKINSON DISEASE. **284**, 15951-15969, doi:10.1074/jbc.M900418200 (2009).
- 5 Ghosh, S., Mahapatra, A. & Chattopadhyay, K. Modulation of α -Synuclein Aggregation by Cytochrome c Binding and Hetero-dityrosine Adduct Formation. *ACS Chemical Neuroscience* **10**, 1300-1310, doi:10.1021/acscemneuro.8b00393 (2019).

- 6 Yang, W., Li, X., Li, X., Li, X. & Yu, S. Neuronal hemoglobin in mitochondria is reduced by forming a complex with α -synuclein in aging monkey brains. *Oncotarget* **7**, 7441-7454, doi:10.18632/oncotarget.7046 (2016).
- 7 Kleinknecht, A. *et al.* C-Terminal Tyrosine Residue Modifications Modulate the Protective Phosphorylation of Serine 129 of α -Synuclein in a Yeast Model of Parkinson's Disease. *PLoS genetics* **12**, e1006098-e1006098, doi:10.1371/journal.pgen.1006098 (2016).
- 8 Bopardikar, M. *et al.* Triphala inhibits alpha-synuclein fibrillization and their interaction study by NMR provides insights into the self-association of the protein. *RSC Advances* **9**, 28470-28477, doi:10.1039/C9RA05551G (2019).
- 9 Hoffmann, A.-C. *et al.* Extracellular aggregated alpha synuclein primarily triggers lysosomal dysfunction in neural cells prevented by trehalose. *Sci Rep* **9**, 544-544, doi:10.1038/s41598-018-35811-8 (2019).
- 10 Fleming, P. J. & Fleming, K. G. HullRad: Fast Calculations of Folded and Disordered Protein and Nucleic Acid Hydrodynamic Properties. *Biophysical Journal* **114**, 856-869, doi:10.1016/j.bpj.2018.01.002 (2018).
- 11 Wilkins, D. K. *et al.* Hydrodynamic radii of native and denatured proteins measured by pulse field gradient NMR techniques. *Biochemistry* **38**, 16424-16431, doi:10.1021/bi991765q (1999).
- 12 Hill, S. E., Robinson, J., Matthews, G. & Muschol, M. Amyloid protofibrils of lysozyme nucleate and grow via oligomer fusion. *Biophysical journal* **96**, 3781-3790, doi:10.1016/j.bpj.2009.01.044 (2009).
- 13 Haugstad, G. (2015).
- 14 Heymann, J. B., Möller, C. & Müller, D. J. Sampling effects influence heights measured with atomic force microscopy. **207**, 43-51, doi:10.1046/j.1365-2818.2002.01039.x (2002).
- 15 Habchi, J. *et al.* An anticancer drug suppresses the primary nucleation reaction that initiates the production of the toxic A β 42 aggregates linked with Alzheimer's disease. *Science Advances* **2**, e1501244, doi:10.1126/sciadv.1501244 (2016).
- 16 Narayanan, V. & Scarlata, S. Membrane Binding and Self-Association of α -Synucleins. *Biochemistry* **40**, 9927-9934, doi:10.1021/bi002952n (2001).
- 17 Osuji, C.
- 18 Boyer, D. *et al.* (bioRxiv, 2019).
- 19 Davidson, W. S., Jonas, A., Clayton, D. F. & George, J. M. Stabilization of alpha-synuclein secondary structure upon binding to synthetic membranes. *J Biol Chem* **273**, 9443-9449, doi:10.1074/jbc.273.16.9443 (1998).
- 20 Lorenzen, N., Lemminger, L., Pedersen, J. N., Nielsen, S. B. & Otzen, D. E. The N-terminus of α -synuclein is essential for both monomeric and oligomeric interactions with membranes. *FEBS letters* **588**, 497-502, doi:10.1016/j.febslet.2013.12.015 (2014).
- 21 van Rooijen, B. D., Claessens, M. M. & Subramaniam, V. Lipid bilayer disruption by oligomeric alpha-synuclein depends on bilayer charge and accessibility of the hydrophobic core. *Biochim. Biophys. Acta, Biomembr.* **1788**, 1271 (2009).
- 22 Volles, M. J. *et al.* Vesicle permeabilization by protofibrillar alpha-synuclein: implications for the pathogenesis and treatment of Parkinson's disease. *Biochemistry* **40**, 7812 (2001).
- 23 Shahane, G., Ding, W., Palaiokostas, M. & Orsi, M. Physical properties of model biological lipid bilayers: insights from all-atom molecular dynamics simulations. *Journal of Molecular Modeling* **25**, 76, doi:10.1007/s00894-019-3964-0 (2019).
- 24 Sastry, P. S. Lipids of nervous tissue: composition and metabolism. *Progress in lipid research* **24**, 69-176, doi:10.1016/0163-7827(85)90011-6 (1985).

Reviewers' comments:

Reviewer #1 (Remarks to the Author):

The revision made by Professor Chattopadhyay to the questions proposed in the previous version of the manuscript have been well answered and argued.

Reviewer #2 (Remarks to the Author):

I remain having reservations about this study.

- The structural interpretation of the low resolution cryo EM density is highly speculative. The authors say "When compared, the density maps of both oligomers1 and oligomers2 show a striking resemblance with a tetrameric model of α -Syn arranged in accordance with the Greek-key pattern". I disagree here. The tetrameric model does not fit well to the density obtained from the cryo-EM analysis and I don't see a reason why to prefer this tetrameric model over other folds. It is a misleading over-interpretation when the authors talk in the abstract about "the Greek key-like architecture of the mace oligomers". The conclusion that there would be conformational distortion just originates from the observation that the greek-key tetramer doesn't really fit. That is not a sufficient basis for suggesting a folding mechanism.

- The topic of aSyn tetramers has been heavily discussed in the literature; when aSyn tetramer formation is suggested it would be much better to complement the SEC with MALS than to omit the SEC data, see Fauvet et al. DOI: 10.1074/jbc.M111.318949

- The negative-stain TEM data is of poor contrast and does not justify the homogeneity statement "a nearly homogeneous distribution of small linear-shaped oligomers (<10 nm) having a distinct shape, following heme treatment at either pre-incubation or post incubation stage". The description of a linear-shaped oligomer moreover does neither fit to the mace density nor to the Greek-key tetramer applied later to fit the electron density.

- The particles observed by AFM fit the cryo-EM density neither in the x/y- nor in the z-dimensions. If the problem in the x/y-dimensions is indeed object-tip convolution, a narrower tip should be applied to obtain higher quality images. Otherwise it remains speculative if the different techniques are investigating the same oligomers.

- The FTIR spectra of oligomer2 and aSyn incubated without heme look similar, but different than the spectrum of oligomer1, in contrast to the conclusions the authors draw from the rather fragile analysis of the content of % antiparallel β -sheet.

- My previous point 5 was addressing the question if the rapid drop in ThT fluorescence after addition of heme is really due to fibril disassembly or rather an effect on the ThT fluorescence of fibrils (not on the ThT fluorescence in the absence of fibrils). Quantification of monomers and oligomers during disassembly would be essential, as the current data does not provide a quantitative description of this process.

- The concluding scheme 7 is highly speculative. Furthermore, I don't think that the title of the manuscript is appropriate because it has neither been shown that the oligomers are fibril-forming nor that there is conformational distortion.

Reviewer #3 (Remarks to the Author):

The revised manuscript "Conformational distortion in a fibril forming oligomer arrests alpha synuclein fibrillation and minimizes its toxic effects" submitted by Ritobrata Chakraborty et al has improved significantly. Figure quality, responses to reviewers, scholarly presentation etc have become much better. The manuscript addressed important question of ASyn oligomers and effect of heme on its structural distortion. Small points may be addressed before final version:

1. Axes label for some figure can be made clear. I believe production editor will take care of this.
2. As discussed by authors, many researchers have shown that ASyn can be isolated in the tetrameric state directly from cells in non-denaturing condition. Here as well, authors proposed tetrameric structure of heme stabilized oligomers. Can there be a structural-toxicity similarity between those two states?
3. Heme binding to AS is in lower micromolar range. I assume that peak disappearance should be evident at that K_d . I don't see much intensity decay here? Are authors confident about K_d value?
4. Several new structure of ASyn fibrils have appeared using cryoEM, which showed distorted kernel. It will be worth discussing few of those in relevance to the model presented here.

In conclusion, I see a much-improved version of the work and hence recommend to be accepted.

Response to the Reviewer 2 and Reviewer 3's comments:

We are thankful to the reviewers for their renewed assessment of our work. We have clarified our results and claims further so that they meet the reviewers' expectations. We believe that our manuscript is now suitable for publication in Communications Biology.

Response to Reviewer 2's Comments

Reviewer #2 (Remarks to the Author):

I remain having reservations about this study. The structural interpretation of the low resolution cryo-EM density is highly speculative.

The resolution of the cryo-EM density map of our heme-treated alpha-Synuclein (α -Syn) oligomers₁ (heme added at time=0) and oligomers₂ (heme added at later time-points) is 10.4 Å and 8.9 Å, respectively, at the 0.143 FSC criterion. Such moderate resolution has resulted from the low molecular weight (MW) of the oligomers (60 kDa, estimated from SEC-MALS, Figure S8, Supporting Information, and Figure R1, below), low net concentration of the individual populations of the oligomers within the multiple aggregation pathways, as well as the rapid structural dynamics of α -Syn, which is a natively unfolded protein. We would like to reiterate that the structural understanding of such low-molecular weight, small-sized oligomers of α -Syn (or any other protein involved in any neurodegenerative diseases) is extremely limited. To the best of our knowledge, the present finding is the first structure of a non-toxic oligomer of α -Syn. The only other structural study available used cryo-EM to report structures at 18 Å and 19 Å (0.5 FSC criterion) resolution of toxic (in contrast to the non-toxic oligomers in the present study) oligomeric forms of α -Syn (Chen SW. et al., PNAS 2015).

We have shown using a molecular dynamics simulation that the non-toxic oligomers (end-products of heme-induced inhibition of fibrillation) would have been on-pathway oligomers if not for the structural distortion near their C terminal, which prevents their appending into fibrils. Also, our study is unique and innovative in that it has identified a stable oligomer of α -Syn, which has been only possible due to the addition of heme which binds to a specific His residue (which we proved using a His50Gln mutant). We believe that this strategy (of blocking the His residue using heme or heme-like small molecules) would enable the solving of high resolution structures using newer generation cryo-EM hardware and algorithms. More importantly, this approach will help develop therapeutic strategies targeting α -Syn aggregates, which has not been possible so far because of the lack of structural data.

Since our cryo-EM data of the heme-treated oligomers are of nanometre resolution, we have validated all important aspects of this structure using additional biochemical and biophysical methods, namely, FT-IR, single molecule fluorescence spectroscopy, native PAGE, site directed mutagenesis and enzyme kinetic assays for heme peroxidase activity. We have also provided detailed toxicity measurements of these oligomers and other aggregated components of the protein using multiple relevant assays. We are happy to mention that the other two reviewers are satisfied with these analyses.

The authors say “When compared, the density maps of both oligomers₁ and oligomers₂ show a striking resemblance with a tetrameric model of α -Syn arranged in accordance with the Greek-key pattern”. I disagree here. The tetrameric model does not fit well to the density obtained from the cryo-EM analysis and I don't see a reason why to prefer

this tetrameric model over other folds. It is a misleading over-interpretation when the authors talk in the abstract about “the Greek key-like architecture of the mace oligomers”. The conclusion that there would be conformational distortion just originates from the observation that the greek-key tetramer doesn’t really fit. That is not a sufficient basis for suggesting a folding mechanism.

We apologize for the misunderstanding. Below, we address the reviewer’s concern. This question has two principal issues, namely ‘Greek key fold’ and the ‘tetrameric structure’ of the oligomers. Hence, we have divided our reply into these two subheadings:

The Greek key:

It is known that α -Syn oligomers/fibrils comprise predominantly a β -strand-rich architecture, as has also been reported in our study. Regarding the structural fold, α -Syn fibril polymorph structures (Figure R1) published so far possess a variation of the ‘Greek key motif’, also termed as the ‘bent β arch’ (Li, Ge et al. 2018) within the kernel or core of the fibril (PDB 2N0A, 6CU7, 6CU8, 6FLT, 6H6B, 6RT0, 6RTB, 6SST and 6SSX). When we stated a ‘resemblance’ between the Greek key topology and the cryo-EM density of heme-treated oligomers, we were referring to the similarity between the two structures in their overall shape and size by visual inspection (a heavy C-terminal head with a rod-shaped base). Further, when we tried to fit the established Greek key structure onto our density map, we found incompatibility between the two. On closer inspection, we observed a ‘distortion’ within the structure of our heme-treated oligomer. We therefore tried to fit the head and base of the established structure (PDB 2N0A) separately within our oligomer density map, and found that they were compatible. Furthermore, when we compared this separately-fitted structure to the established structure of 2N0A as well as the other models, we noted a ‘contortion’ between the head and base of the structure.

Figure R1: Some existing α -Syn fibril Greek key/bent β -arch kernel structures showing a heavy head with a base feature.

The end product of α -Syn aggregation typically leads to fibril formation, which are either rod-shaped or twisted polymorphs (Li, Ge et al. 2018). In our study the end products of aggregation of WT α -Syn are typically rod-shaped fibrils (Figure R2, negative-stain TEM micrograph shows the presence of straight rod-shaped fibrils). These rod fibril polymorphs (whose high-resolution structures have been solved by many groups before) are composed primarily of a Greek key (Tuttle, Comellas et al. 2016) (PDB 2N0A) architecture (Guerrero-Ferreira, Taylor et al. 2018, Li, Ge et al. 2018, Guerrero-Ferreira, Kovacic et al. 2020), and that is the reason we used this fold in our study in our comparison with the density maps of heme-treated oligomers (that were also prepared using WT α -Syn). We can understand that the Reviewer 2 may have a concern with the term ‘Greek key fold’, but without a specific suggestion of another relevant fold (available neither in his/her report nor in existing literature), this model had to be

our best scientifically-relevant starting point, which did fit to our density maps (Figure R3).

Figure R2: Negative-stain TEM micrograph showing the ‘rod’-like morphology of WT α -Syn fibrils. The rod fibril polymorph is known to have a longer pitch of 92 nm while the ‘twister’ polymorph (found predominantly in mutant α -Syn fibrils) has a short pitch of 46 nm (not observed in any of the fibrils in the above micrograph).

Figure R3: (A) A representation of four α -Syn chains arranged as per the Greek key architecture (PDB 2N0A) in cartoon representation (blue) inside a semi-transparent surface. Only the structured part (residue 37-99) has been included. (B) The complete model shown in Figure A could not be fitted into the density map of oligomer₁ as a rigid piece. Thus, the ‘head’ and ‘base’ parts of the model from Figure A were fitted separately (as orange CPK sphere representations) into the density map generated from oligomers₁ (semi-transparent orange envelope). (C) Superimposition of the Greek key model from Figure A (in blue) and the fitted model from Figure B (in orange), both represented as spheres, shows distortion in the fitted model near its ‘head’- ‘base’ junction.

Figure R4: The well-equilibrated and stable end-simulation structure that was previously docked with heme (A, in red), shows the presence of a twist/ distortion that matches with the cryo-EM fitted model of 2N0A (B, in green). The protofilament kernels (residues 43-83) of rod (6CU7) and twister (6CU8) polymorphs show structural differences (Li B. et al; Nat. Commun. 2018). Interestingly, when we compared 6CU7 and 6CU8 (C) we found a tendency of distortion at the head and base junction.

Tetrameric Structure:

We have shown that the Greek key tetrameric model fits well within the density maps of the heme-stabilized oligomers. Since our structure is of nanometre resolution, we used additional biophysical and biochemical methods to compliment the tetrameric composition. First, native PAGE data clearly show that the oligomers₁ and oligomers₂ (formed by adding heme either at the beginning or at the end of aggregation) are tetrameric (Figure R5, and Figure S8, SI).

Figure R5: A 10 % Native PAGE of AlexaFluor488-tagged α -Syn monomer (MW ~15 kDa, first lane from left), 5-TAMRA, SE-tagged horseradish peroxidase (HRP, MW ~44 kDa, lane 2), AlexaFluor488-tagged α -Syn oligomers₁ (lane 3), AlexaFluor488-tagged α -Syn oligomers₂ (lane 4), 5-TAMRA, SE-tagged Bovine Serum Albumin (BSA, 66.4 kDa, lane 5). The monomer migrated at ~15 kDa as has been reported before, while the oligomers₁ and oligomers₂ migrated at ~60 kDa. These fluorescently labeled proteins were also subjected to FCS to correlate the FCS r_H values with their observed migration pattern.

Second, we carried out single molecule fluorescence correlation spectroscopy to obtain the hydrodynamic radius of the heme-treated oligomers and showed that this value completely matches the value for the radius of gyration of a model tetrameric oligomer obtained using computational modelling using an independent algorithm (main text).

Finally, we have now provided SEC-MALS data which clearly suggest that oligomer₁ is a tetramer (Figure R6, and Figure S8, SI).

The topic of aSyn tetramers has been heavily discussed in the literature; when aSyn tetramer formation is suggested it would be much better to complement the SEC with MALS than to omit the SEC data, see Fauvet et al. DOI: 10.1074/jbc.M111.318949

We have now complemented our native PAGE and FCS data (that suggest that the heme-treated oligomers are indeed tetrameric) with molar mass determination using SEC-MALS (refer page 13 and 23, main text) as insisted by R2. The analysis has been depicted in Figure R6, and Figure S8, SI. The data from light scattering and differential refractive index detectors reveal the molar mass for the oligomer₁ as 60.2 kDa. This was compared with BSA (control) that showed a molar mass of 66.4 kDa. The molar mass of monomeric protein is estimated to be 14.5 kDa using SEC-MALS, as has been established previously (Fauvet B et al., J. Biol. Chem., 2012).

Figure R6: SEC-MALS analyses of Bovine Serum Albumin (BSA, 66.4 g/mol, in black), monomeric α -Synuclein (14.46 g/mol, in blue) and heme-treated oligomers₁ (60.2 g/mol, in red). The continuous lines correspond to the differential refractive indices (660 nm, left ordinate axis), and horizontal line segments correspond to the calculated molar masses for the corresponding peaks (right ordinate axis).

The tetrameric structure of α -Syn previously studied (Bartels, Choi et al. 2011) was for ‘non fibrillating native structure’ and the proposed fold was α -helical. However, our oligomers are ‘non-fibrillating’ because of the presence of heme.

It may be noted that in addition to the SEC-MALS, we have provided substantial data in the form of a native PAGE (Figure R5, and Figure S9, SI) to gain insight into the apparent MW of the heme-treated oligomers. The data very clearly show that the monomer (Lane 1) and our heme treated oligomers (Lane 3: oligomers₁ and Lane 4: oligomers₂) have very different migration distances from their respective wells, proving that our oligomers and the monomers have dissimilar MW. Complimentary to established native PAGE analyses (Chen SW. et al., PNAS, 2015, Bopardikar M et al., RSC Adv., 2019, Hoffmann A. et al., Sci. Rep., 2019) the α -Syn monomer migrated with a MW of ~15 kDa. The oligomers₁ and oligomers₂ migrate with a MW of ~ 60 kDa (when compared to the migration patterns of two reference proteins: Bovine Serum Albumin, 66.4 kDa and Horseradish peroxidase, 44 kDa).

To support this claim further, we estimated the theoretical hydrodynamic radius ($r_{H, \text{Theo}}$) of a tetrameric model of α -Syn obtained from its PDB structure (ID 2N0A) using the HullRad algorithm (Fleming P. et al., Biophys. J., 2018). We then compared it with the experimental $r_{H, \text{Exp}}$ of the heme-treated oligomers calculated using fluorescence correlation spectroscopy or FCS (Figure R5, below). We found that the theoretical $r_{H, \text{Theo}}$ value was in excellent agreement with the experimental $r_{H, \text{Exp}}$. All these results are substantial in proving that the heme-induced oligomers are indeed tetrameric.

The negative-stain TEM data is of poor contrast and does not justify the homogeneity statement “a nearly homogeneous distribution of small linear-shaped oligomers (<10 nm) having a distinct shape, following heme treatment at either pre-incubation or post incubation stage”. The description of a linear-shaped oligomer moreover does neither fit to the mace density nor to the Greek-key tetramer applied later to fit the electron density.

We respectfully disagree. We have shown below an example, using AFM, of the heterogeneous nature of the untreated aggregation landscape of α -Syn and the near-homogeneous populations of oligomer₁ and oligomer₂, which were obtained by the addition of heme (Figure R7).

Figure R7: AFM micrographs show that heme addition to structurally-heterogeneous populations of α Syn aggregates after \sim 48 hours of aggregation (left panel) converts them into a nearly-homogeneous array of oligomers (right panel).

The particles observed by AFM fit the cryo-EM density neither in the x/y- nor in the z dimensions. If the problem in the x/y-dimensions is indeed object-tip convolution, a narrower tip should be applied to obtain higher quality images. Otherwise it remains speculative if the different techniques are investigating the same oligomers.

We respectfully disagree. We do not think that AFM ‘diameter’ data can be used to match TEM diameter data. This is well known and already published.

The following paragraph explains this matter:

An AFM image is a convolution of the shape of the probe and the shape of the sample. In AFM, the lateral resolution is low (\sim 30nm) due to this convolution, while the vertical resolution can be up to 0.1 nm. This makes protruding features to appear wide, while holes appear smaller (both narrower and often less deep, too). Less sharp probes can enhance this effect. Thus, when the dimensions of an object are measured on AFM using the semi-contact method in air, an inflated estimate of the object width and an underestimated height of the object are obtained, which introduces an error which in the parameters of the measured objects (Figure R8, below)

Figure R8: Dilation of lateral particle dimension during AFM imaging: The finite size of the scanning tip, shown here as a parabola, increases the apparent width (W_{app}) of the object well beyond its actual width (W). For an ellipse, W can be derived from the measured height H and W_{app} of the object and the tip radius R_t (Hill, Robinson et al. 2009).

Therefore, due to the aforementioned limitation, the lateral width of the object appears larger than its actual diameter. Hence, the height of the particles (in this case oligomers and fibrils) are usually reported (Heymann, Möller et al. 2002, Haugstad 2015) instead of the width (refer AFM height histograms, Figure 1 B, C, F, main text). We have found that our values match very well with the established reports on the AFM height as well as TEM diameter of the α Syn aggregates.

It may be noted that in all versions of this manuscript, we have never compared the diameters between AFM and TEM. Instead, TEM data were used for the diameter information, while AFM data were used to measure the heights, as they should be.

The FTIR spectra of oligomer2 and aSyn incubated without heme look similar, but different than the spectrum of oligomer1, in contrast to the conclusions the authors draw from the rather fragile analysis of the content of % antiparallel β -sheet.

The Reviewer 2 is requested to note that the second derivative spectra of heme-treated oligomers lacks completely the minima at 1695 cm^{-1} , which is the defining feature of anti parallel β sheet structure. In the absence of any specific suggestion or technical details, it is difficult for us to update our fitting routines. Since it is difficult (or even impossible) to determine the secondary structure content using just a visual observation of the FTIR spectrum, we have provided detailed quantitative analyses routines for the data and none of the other reviewers had any specific complaints to that procedure. It may also be noted this method of quantification of FT-IR data has been already published (Chen, Drakulic et al. 2015, Chowdhury, Sen et al. 2019) and followed by several other groups.

My previous point 5 was addressing the question if the rapid drop in ThT fluorescence after addition of heme is really due to fibril disassembly or rather an effect on the ThT fluorescence of fibrils (not on the ThT fluorescence in the absence of fibrils). Quantification of monomers and oligomers during disassembly would be essential, as the current data does not provide a quantitative description of this process.

We have shown that ThT fluorescence is not changed by adding heme at the concentration of our study (Figure R9).

Figure R9: The concentration of heme (hemin chloride) used in our studies (which is 8 μM) does not have a quenching effect on the fluorescence of 20 μM Thioflavin T. Both hemin chloride and ThT were dissolved in sodium phosphate buffer, pH 7.5.

In an unlikely but hypothetical situation, if the fluorescence of ThT bound to $\alpha\text{-Syn}$ fibrils behaves very differently from the ThT fluorescence in solution due to heme addition, it may be noted that fluorescence quenching (heme quenching of the ThT fluorescence of fibrils) should always occur in the nsec or psec timescale. It can sometimes occur in the microsec time scale for few large lifetime lanthanides etc., but never in the time range of minutes and hours (which is the time scale of fluorescence decrease which was observed for the heme-induced disassembly process as reported here (Figure R10)).

Figure R10: The Thioflavin T fluorescence of higher order aggregates (fibrils) of $\alpha\text{-Syn}$ shows a decrease upon addition of heme. Note that this process occurs in the matter of minutes (and not in the ns-ps range, which would have corresponded to ThT fluorescence quenching by heme).

Most importantly, the heme induced disassembly of the fibrils has been cleanly probed by direct observation using TEM and AFM (Figure R11). Since these two techniques do not require ThT as a probe, its effect on heme addition (as mentioned by the Reviewer 2) does not hold here.

Figure R11: Conversion of higher-order fibrillar aggregates (left panel) of α -Syn into oligomers₁ (middle panel) and oligomers₂ (right panel), as observed using AFM and negative-stain TEM, supports our finding that it is indeed the breakdown of the fibrils that causes a decrease in ThT fluorescence, rather than heme-induced quenching of ThT in the presence of fibrils.

In addition, we have now provided the real-time NMR data for the fibrillar breakdown induced upon addition of heme (Figure R12). The subsequent change in the spectrum is also not affected by ThT as the NMR sample solution did not contain any ThT.

Figure R12: Selected 1D ^1H NMR spectra for 200 μM α -synuclein fibrils (control for protein only in the lowest spectrum) in the presence of 8 μM hemin chloride collected over time. The representation shows the appearance of new peaks over time- shaded in green and the decay of specific peaks- shaded in grey. The time-dependent changes in the intensity are represented for the ^1H NMR peak resonances between 8 ppm and \sim 6.7 ppm and around 1.1 and 0.2 ppm.

Our 1D ^1H NMR experiments performed on mature fibrils incubated with hemin at 37°C showed:

- 1) an immediate reduction of aromatic peak intensities accompanied by a slight up-field chemical shift perturbations observed for a few peaks in this region (Figure R12, left panel), indicating involvement of aromatic interactions in mediating this hetero-molecular association with hemin chloride;

- 2) new signals with gradually-improving S/N ratio that were observed between t=26 h and 42 h (Figure S12, middle panel). Major changes were observed for the chemical shift resonances in the aliphatic region of the fibrillar spectra, suggesting a direct molecular interaction of heme chloride with the side chain protons- prompting them to be opened from the fibrillar orientation. Increasing intensities for the new peaks with sharp signal over time indicated a heme-induced shift in equilibrium between soluble and insoluble oligomeric conformers, more towards the soluble species. Thus, heme modulates side-chain dynamics, eventually arresting the protein conformers in a soluble intermediate that suspends further oligomerization events;
- 3) a new peak appears after 8 h (and intensifies at 72 h of incubation) at ~0.17 ppm (solvent protected alkyl chain region) further reinstating the altered mobility and solvent exposure upon molecular interaction with heme chloride. Alternatively, the appearance of this peak indicates the formation of large spherical oligomeric conformers of ~50 nm that have been extensively reported in literature for several studied amyloidogenic proteins (Ratha BN. et al., Proteins: Struct. Func. Bioinformatics, Weiss MA., Vitam Horm. 2009, Huang R. et al., J. Mol. Biol. 2012).

Collectively, we have been able to support the ThT fluorescence, TEM and AFM data with the results discussed above.

However, we are not clear about what the R2 means by quantification of structural species formed due to fibril disassembly. Maybe, he wants to determine how the concentrations of different species (fibrils/oligomers/monomers etc.) change as a result of heme-induced fibril disassembly. The process of fibril disassembly into oligomers₂ in presence of heme is more heterogeneous than the onward process of oligomer₁ formation. Therefore, the presence of even a very small fraction of large-sized particles along with the majority of oligomers₂ formed as a result of fibril disassembly will be detected by the SEC-MALS light scattering detectors. Most importantly, SEC-MALS data on the heterogeneous, polydisperse landscape of proteins, which has never been reported as far per our knowledge, is not even relevant in this study. If we could figure out how different populations of oligomers/ monomers evolve in the process of fibril disintegration, this would completely shift the focus from an important paper: in which we are reporting, for the very first time, the cryo-EM reconstructed structure of non-toxic oligomers of α -Syn.

In addition to these new data, we have toned down our claims in several places of the manuscript.

The concluding scheme 7 is highly speculative. Furthermore, I don't think that the title of the manuscript is appropriate because it has neither been shown that the oligomers are fibril-forming nor that there is conformational distortion.

We sincerely hope that the reviewer will change his/her opinion in light of the new data provided in this version.

Perhaps, if the editors agree, we would propose a slightly different title as follows:

Heme-induced conformational distortion in a fibril-forming oligomer arrests alpha-Synuclein fibrillation and minimizes its toxic effects

Response to Reviewer 3's Comments

Reviewer #3 (Remarks to the Author):

The revised manuscript "Conformational distortion in a fibril forming oligomer arrests alpha synuclein fibrillation and minimizes its toxic effects" submitted by Ritabrata Chakraborty et al has improved significantly. Figure quality, responses to reviewers, scholarly presentation etc have become much better. The manuscript addressed important question of ASyn oligomers and effect of heme on its structural distortion. Small points may be addressed before final version:

We are extremely grateful to Reviewer 3 for his/ her constructive assessment of our manuscript. Following his/ her suggestions, we have made some modifications that are mentioned below:

Axes label for some figure can be made clear. I believe production editor will take care of this.

We have now made the axes labels of all the figures more legible for our readers.

As discussed by authors, many researchers have shown that ASyn can be isolated in the tetrameric state directly from cells in non-denaturing condition. Here as well, authors proposed tetrameric structure of heme stabilized oligomers. Can there be a structural-toxicity similarity between those two states?

The Reviewer 3 is correct in pointing out the similarities between the tetrameric natively purified cell-derived α -Syn (Bartels, Choi et al. 2011, Wang, Perovic et al. 2011) with our heme-stabilized tetramer. Although both structures have been found to be tetrameric and non aggregating, a major difference in the two lies in their secondary structure content. The cell-derived structure is predominantly α -helical, while we have shown that our oligomer comprises β sheet architecture. Moreover, we have hypothesized that an undistorted version of our β sheet tetramer is an early-stage 'primary' oligomer, which appends with more tetramers to form protofilaments that subsequently intertwine at the steric zipper interface (Li, Ge et al. 2018) to form mature fibrils (9-10 nm wide). Heme interaction to His50 distorts this primary tetramer, thereby preventing protofilament and fibril formation. We are now trying to understand whether heme interacts with α -Syn inside cells leading to the formation of heme treated α -Syn oligomers. We have also made plans to study their structure *in situ* using cryoelectron tomography.

Heme binding to AS is in lower micromolar range. I assume that peak disappearance should be evident at that Kd. I do not see much intensity decay here? Are authors confident about Kd value?

We are grateful to the reviewer for his keen eye. We request him to note the Figure R13, wherein we have shown the peak disappearance of TMR- α -Syn showing a rapid and steep decrease in the intensity upon the addition of minute concentrations of heme.

Figure R13: Fluorescence quenching due to binding of α -Syn-TMR by heme.

Moreover, we would like to add that our 1D NMR (Figure S12, SI) study on the binding between heme and α -Syn has provided us with an apparent K_D of 0.413 μ M which is very much comparable with our fluorescence quenching-based experimental K_D of 0.591 μ M.

Several new structure of Asyn fibrils have appeared using cryoEM, which showed distorted kernel. It will be worth discussing few of those in relevance to the model presented here.

We are grateful to the reviewer for this excellent suggestion. We compared the available structures of the α -Syn kernel, and found that the difference is primarily localized around the His50 residue (near the head-base junction of the structure). We have pointed out this feature in the discussion now.

Figure R14: PDB codes: 6CU7, 6CU8, 6H6B, 6FLT

In conclusion, I see a much-improved version of the work and hence recommend it to be accepted.

We sincerely thank the reviewer for the positive assessment of our work.

References:

- Bartels, T., J. G. Choi and D. J. Selkoe (2011). " α -Synuclein occurs physiologically as a helically folded tetramer that resists aggregation." *Nature* 477(7362): 107-110. Bopardikar, M., A. Bhattacharya, V. M. Rao Kakita, K. Rachineni, L. C. Borde, S. Choudhary, S. R. Koti Ainaravapu and R. V. Hosur (2019). "Triphala inhibits alpha-synuclein fibrillization and their interaction study by NMR provides insight s into the self-association of the protein." *RSC Advances* 9(49): 28470-28477.
- Chen, S. W., S. Drakulic, E. Deas, M. Ouberai, F. A. Aprile, R. Arranz, S. Ness, C. Roodveldt, T. Williams, E. J. De-Genst, D. Klenerman, N. W. Wood, T. P. J. Knowles, C. Alfonso, G. Rivas, A. Y. Abramov, J. M. Valpuesta, C. M. Dobson and N. Cremades (2015). "Structural characterization of toxic oligomers that are kinetically trapped during α -synuclein fibril formation." *Proc. Natl. Acad. Sci. U. S. A.* 112: E1994.
- Chowdhury, S., S. Sen, A. Banerjee, V. N. Uversky, U. Maulik and K. Chattopadhyay (2019). "Network mapping of the conformational heterogeneity of SOD1 by deploying statistical cluster analysis of FTIR spectra." *Cellular and Molecular Life Sciences* 76(20): 4145-4154.
- Fleming, P. J. and K. G. Fleming (2018). "HullRad: Fast Calculations of Folded and Disordered Protein and Nucleic Acid Hydrodynamic Properties." *Biophysical Journal* 114(4): 856-869.
- Guerrero-Ferreira, R., L. Kovacic, D. Ni and H. Stahlberg (2020). "New insights on the structure of alpha-synuclein fibrils using cryo-electron microscopy." *Curr Opin Neurobiol* 61: 89-95.
- Guerrero-Ferreira, R., N. M. Taylor, D. Mona, P. Ringler, M. E. Lauer, R. Riek, M. Britschgi and H. Stahlberg (2018). "Cryo-EM structure of alpha-synuclein fibrils." *Elife* 7. Haugstad, G. (2015). Overview of Atomic Force Microscopy.
- Heymann, J. B., C. Möller and D. J. Müller (2002). "Sampling effects influence heights measured with atomic force microscopy." 207(1): 43-51.
- Hill, S. E., J. Robinson, G. Matthews and M. Muschol (2009). "Amyloid protofibrils of lysozyme nucleate and grow via oligomer fusion." *Biophysical journal* 96(9): 3781-3790.
- Hoffmann, A.-C., G. Minakaki, S. Menges, R. Salvi, S. Savitskiy, A. Kazman, H. Vicente Miranda, D. Mielenz, J. Klucken, J. Winkler and W. Xiang (2019). "Extracellular aggregated alpha synuclein primarily triggers lysosomal dysfunction in neural cells prevented by trehalose." *Scientific reports* 9(1): 544-544.
- Li, B., P. Ge, K. A. Murray, P. Sheth, M. Zhang, G. Nair, M. R. Sawaya, W. S. Shin, D. R. Boyer, S. Ye, D. S. Eisenberg, Z. H. Zhou and L. Jiang (2018). "Cryo-EM of full-length α synuclein reveals fibril polymorphs with a common structural kernel." *Nature Communications* 9(1): 3609.
- Tuttle, M. D., G. Comellas, A. J. Nieuwkoop, D. J. Covell, D. A. Berthold, K. D. Kloepper, J. M. Courtney, J. K. Kim, A. M. Barclay, A. Kendall, W. Wan, G. Stubbs, C. D. Schwieters, V. M. Lee, J. M. George and C. M. Rienstra (2016). "Solid-state NMR structure of a pathogenic fibril of full-length human alpha-synuclein." *Nat Struct Mol Biol* 23(5): 409-415. Wang, W., I. Perovic, J. Chittuluru, A. Kaganovich, L. T. Nguyen, J. Liao, J. R. Auclair, D. Johnson, A. Landeru, A. K. Simorellis, S. Ju, M. R. Cookson, F. J. Asturias, J. N. Agar, B. N. Webb, C. Kang, D. Ringe, G. A. Petsko, T. C. Pochapsky and Q. Q. Hoang (2011). "A soluble α -synuclein construct forms a dynamic tetramer." *Proc Natl Acad Sci U S A* 108(43): 17797-17802.
- Wilkins, D. K., S. B. Grimshaw, V. Receveur, C. M. Dobson, J. A. Jones and L. J. Smith (1999). "Hydrodynamic radii of native and denatured proteins measured by pulse field

gradient NMR techniques." Biochemistry 38(50): 16424-16431.

REVIEWERS' COMMENTS:

Reviewer #2 (Remarks to the Author):

I thank the authors for their efforts in replying to my comments. The SEC-MALS data is convincing and valuable. I like the suggested modified, more specific, title, but the authors should choose their preferred title. I recommend publication.